# Spatial inter-centromeric interactions facilitated the emergence of evolutionary new centromeres

Krishnendu Guin[1], Yao Chen[2], Radha Mishra[1†], Siti Rawaidah BM Muzaki[2], Bhagya C Thimmappa[1‡], Caoimhe E O'Brien[3], Geraldine Butler[3], Amartya Sanyal[2]*, Kaustuv Sanyal[1]*

[1]Molecular Mycology Laboratory, Molecular Biology and Genetics Unit, Jawaharlal Nehru Centre for Advanced Scientific Research, Bangalore, India; [2]School of Biological Sciences, Nanyang Technological University, Singapore, Singapore; [3]School Of Biomolecular & Biomed Science, Conway Institute of Biomolecular and Biomedical Research, University College Dublin, Dublin, Ireland

*For correspondence:
asanyal@ntu.edu.sg (AS);
sanyal@jncasr.ac.in (KS)

Present address: †Department of Cellular and Molecular Medicine, University of Ottawa, Ottawa, Canada; ‡Department of Biochemistry, Robert-Cedergren Centre of Bioinformatics and Genomics, University of Montreal, Montreal, Canada

Competing interests: The authors declare that no competing interests exist.

**Abstract** Centromeres of *Candida albicans* form on unique and different DNA sequences but a closely related species, *Candida tropicalis*, possesses homogenized inverted repeat (HIR)-associated centromeres. To investigate the mechanism of centromere type transition, we improved the fragmented genome assembly and constructed a chromosome-level genome assembly of *C. tropicalis* by employing PacBio sequencing, chromosome conformation capture sequencing (3C-seq), chromoblot, and genetic analysis of engineered aneuploid strains. Further, we analyzed the 3D genome organization using 3C-seq data, which revealed spatial proximity among the centromeres as well as telomeres of seven chromosomes in *C. tropicalis*. Intriguingly, we observed evidence of inter-centromeric translocations in the common ancestor of *C. albicans* and *C. tropicalis*. Identification of putative centromeres in closely related *Candida sojae*, *Candida viswanathii* and *Candida parapsilosis* indicates loss of ancestral HIR-associated centromeres and establishment of evolutionary new centromeres (ENCs) in *C. albicans*. We propose that spatial proximity of the homologous centromere DNA sequences facilitated karyotype rearrangements and centromere type transitions in human pathogenic yeasts of the CUG-Ser1 clade.

## Introduction

The efficient maintenance of the genetic material and its propagation to subsequent generations determine the fitness of an organism. Genomic rearrangements are often associated with the development of multiple diseases, including cancer. Chromosomal rearrangements, on the other hand, are often observed during speciation (*Searle, 1998*). Such structural changes begin with the formation of at least one DNA double-strand break (DSB), which is generally repaired by homologous recombination (HR) or non-homologous end joining (NHEJ) in vivo. Studies using engineered in vivo model systems suggested that the success of DSB repair through HR depends upon an efficient identification of a template donor. This process of 'homology search' is facilitated by the physical proximity and the extent of DNA sequence homology (*Lee et al., 2016*; *Agmon et al., 2013*; *Burgess and Kleckner, 1999*). Multi-invasion-induced rearrangements (MIRs) involving more than one template donors have recently been shown to be influenced by physical proximity and homology (*Piazza et al., 2017*). Therefore, the nature of genomic rearrangements is mostly dependent on the type of spatial genome organization. In yeasts, apicomplexans, and certain plants, centromeres cluster inside the nucleus (*Muller et al., 2019*), which may facilitate translocations between two chromosomes involving their centromeric and adjacent pericentromeric loci.

The centromere, one of the guardians of genome stability, assembles a large DNA-protein complex to form the kinetochore, which ensures fidelity of chromosome segregation by correctly attaching chromosomes to the spindle. Paradoxically, this conserved process of chromosome segregation is carried out by highly diverse species-specific centromere DNA sequences. For example, the length of centromere DNA is ~125 bp in budding yeast *Saccharomyces cerevisiae* (*Clarke and Carbon, 1980*), but it can be as long as a few megabases in humans (*Mahtani and Willard, 1990*). Centromeres have been cloned and characterized from a large number of fungal species. The only factor that remains common to most fungal centromeres is the presence of histone H3 variant CENP-A$^{Cse4}$ except in some Mucorales like *Mucor circinelloides* (*Navarro-Mendoza et al., 2019*). Many kinetochore proteins are believed to have evolved from pre-eukaryotic lineages and remained conserved within closely related species complexes or expanded through gene duplication (*Meraldi et al., 2006*; *Tromer et al., 2019*; *van Hooff et al., 2017*). It remains a paradox that despite the rapid evolution of centromere DNA, the kinetochore structure remains relatively well-conserved (*Ekwall, 2007*). Therefore, an examination of the evolutionary processes driving species-specific changes in centromere DNA is essential for a better understanding of centromere biology.

The first cloned centromere that of the budding yeast *S. cerevisiae* carries conserved genetic elements capable of forming a functional centromere de novo when cloned into a yeast replicative plasmid (*Clarke and Carbon, 1980*). Such genetic regulation of centromere function also exists in the fission yeast *Schizosaccharomyces pombe,* where centromeres possess inverted repeat-associated structures of 40–100 kb (*Clarke and Baum, 1990*). Other closely related budding and fission yeasts were also found to harbor a DNA sequence-dependent regulation of centromere function (*Gordon et al., 2011*; *Tong et al., 2019*; *Kobayashi et al., 2015*), but the advantage of having such genetic regulation is not well understood. In fact, the majority of species with known centromeres are thought to be regulated by an epigenetic mechanism (*Ekwall, 2007*). A truly epigenetically-regulated fungal centromere carrying a 3–5 kb long CENP-A$^{Cse4}$-bound unique DNA sequence exists in another budding yeast *C. albicans* (*Sanyal et al., 2004*), a CUG-Ser1 clade species in the fungal phylum of Ascomycota. Subsequently, such unique centromeres were also discovered in closely related *Candida dubliniensis* (*Padmanabhan et al., 2008*) and *Candida lusitaniae* (*Kapoor et al., 2015*). Strikingly, all seven centromeres of *C. tropicalis,* another CUG-Ser1 clade species, carry 3–4 kb long inverted repeats (IR) flanking ~3 kb long CENP-A$^{Cse4}$ rich central core (CC). The centromere sequences are highly identical to each other in *C. tropicalis*. Intriguingly, centromere DNA of *C. tropicalis* can facilitate de novo recruitment of CENP-A$^{Cse4}$ to some extent (*Chatterjee et al., 2016*). In contrast, centromeres of *C. albicans* completely lack such a DNA sequence-dependent mechanism (*Baum et al., 2006*). Such a rapid transition in the structural and functional properties of centromeres within two closely related species offers a unique opportunity to study the process of centromere type transition.

Kinetochore proteins appeared as a single punctum at the periphery of a nucleus indicating the presence of constitutively clustered centromeres in *C. tropicalis* (*Chatterjee et al., 2016*). Our previous analysis also showed that centromeres of *C. tropicalis* were located near interchromosomal synteny breakpoints (ICSBs) as relics of ancient translocations in the common ancestor of *C. tropicalis* and *C. albicans* (*Chatterjee et al., 2016*). Do homologous centromere DNA regions in close spatial proximity facilitate chromosomal translocation events? Due to the nature of the then-available fragmented genome assembly, the genome-wide distribution of the ICSBs and the spatial organization of the genome in *C. tropicalis* remained unexplored. However, the near-complete *C. albicans* genome assembly was available. Therefore, to examine whether the spatial proximity of clustered centromeres drives interchromosomal translocation events guiding speciation in the CUG-Ser1 clade required a chromosome-level complete genome assembly of *C. tropicalis*.

In this study, we constructed a chromosome-level gapless genome assembly of the *C. tropicalis* type strain MYA-3404 by combining information from previously available contigs, NGS reads and high-throughput 3C-seq data. Using this assembly and 3C-seq data, we studied the spatial genome organization in *C. tropicalis*. Next, we mapped the ICSBs in the *C. tropicalis* genome with reference to that of *C. albicans* (ASM18296v3) to test whether the frequency of ICSB correlated with the spatial genome organization. In addition, we performed Oxford Nanopore and Illumina sequencing and assembled the genome of *Candida sojae* (strain NCYC-2607), a sister species of *C. tropicalis* in the CUG-Ser1 clade (*Shen et al., 2018*). Finally, using this genome assembly of *C. sojae* and publicly available genome assembly of *C. viswanathii* (ASM332773v1), we identified the putative centromeres

of these two species as HIR-associated loci syntenic to the centromeres of *C. tropicalis*. Based on our results, we propose a model that suggests homology and proximity guided centromere-proximal translocations facilitated karyotype evolution and possibly aided in rapid transition from HIR-associated to unique centromere types in the members of the CUG-Ser1 clade.

## Results

### A chromosome-level gapless assembly of the *C. tropicalis* genome in seven chromosomes

*C. tropicalis* has seven pairs of chromosomes (*Chatterjee et al., 2016*; *Butler et al., 2009*). However, the current publicly available genome assembly (ASM633v3) has 23 nuclear contigs and one mitochondrial contig. To completely assemble the nuclear genome of *C. tropicalis* in seven chromosomes, we combined results of short-read Illumina sequencing and long-read single molecule real-time sequencing (SMRT-seq) with high-throughput 3C-seq (simplified Hi-C) experiment (*Figure 1A*, *Figure 1—figure supplement 1A–D*; *Sexton et al., 2012*). We started from the publicly available genome assembly of *C. tropicalis* strain MYA-3404 in 23 nuclear contigs (ASM633v3, Assembly A) (*Butler et al., 2009*). We used Illumina sequencing reads to scaffold them into 16 contigs to get Assembly B (*Figure 1A*). Next, we used the SMRT-seq long reads to join these contigs, which resulted in an assembly of 12 contigs (Assembly C, *Supplementary file 1*). Based on the contour clamped homogenized electric field (CHEF)-gel karyotyping (*Figure 1B*) and 3C-seq data (*Figure 1—figure supplement 1E–G*), we joined two contigs and rectified a misjoin in Assembly C to produce an assembly of seven chromosomes and five short orphan haplotigs (OHs). We suspected that the OHs are heterozygous loci in the diploid genome of *C. tropicalis*. Analysis of the de novo contigs (*Figure 1—figure supplement 1H*, Materials and methods), sequence coverage data (*Figure 1—figure supplement 2A–B*), and Southern hybridization of engineered aneuploid strains demonstrated that the small OHs mapped to heterozygous regions of the genome (*Figure 1—figure supplement 2C–I*, Materials and methods). Next, we used de novo contigs to fill pre-existing 104 N-gaps and scaffolded 14 sub-telomeres (*Figure 1—figure supplement 3A–C*, *Supplementary file 2*). Finally, we used 3C-seq reads to polish the complete genome assembly of *C. tropicalis* constituting 14,609,527 bp in seven telomere-to-telomere long gapless chromosomes (*Figure 1B*). We call this new assembly as Assembly2020.

We assigned the numbers to each chromosome according to the length, starting from the longest as chromosome 1 (Chr1) through the shortest as chromosome 6 (Chr6). The remaining chromosome, the one containing the rDNA locus, was named as chromosome R (ChrR) (*Figure 1C*). Accordingly, centromeres on each chromosome were named after the respective chromosome number. Additionally, we oriented the DNA sequence of each chromosome in a way to consistently maintain the short arm at the 5′ end. The statistics of these genome assemblies of *C. tropicalis* is summarized in *Supplementary file 3*. In Assembly2020, 1278 out of 1315 Ascomycota-specific BUSCO gene sets could be identified compared to 1255 identified using Assembly A (*Supplementary file 4*, Materials and methods). The inclusion of 23 additional BUSCO gene sets suggests significantly improved contiguity and completeness of Assembly2020.

Previously, using centromere-proximal probes, we could distinctly identify five chromosomes (Chr1, Chr2, Chr3, Chr5, and Chr6) in chromoblot analysis (*Chatterjee et al., 2016*). However, the lengths of Chr4 and ChrR could not be determined. To validate the correct assembly of these two chromosomes (Chr4 and ChrR), we performed additional chromoblot analysis. We observed that Chr4 homologs differed in size (*Figure 1—figure supplement 4A*). Analysis of the sequence coverage across Chr4 identified an internal duplication of ~235 kb region, which could explain the size difference between the homologs Chr4A and Chr4B (*Figure 1C*, *Figure 1—figure supplement 4B*). We named this duplicated locus as *DUP4*. Subsequently, we scanned the entire genome for the presence of copy number variations (CNVs), which led to the identification of two additional large-scale duplication events: one each on Chr5 (*DUP5*,~23 kb) and ChrR (*DUPR*,~80 kb) (*Figure 1C*, *Figure 1—figure supplement 4B*). Further, using CNAtra software (*Khalil et al., 2020*) we confirmed these duplication events and identified additional small-scale CNV loci with copy number <1.5 or >2.5 (*Figure 1—figure supplement 4C*). Additionally, we detected a balanced heterozygous translocation event between Chr1 and Chr4 (*Figure 1—figure supplement 5A*) through analyses of

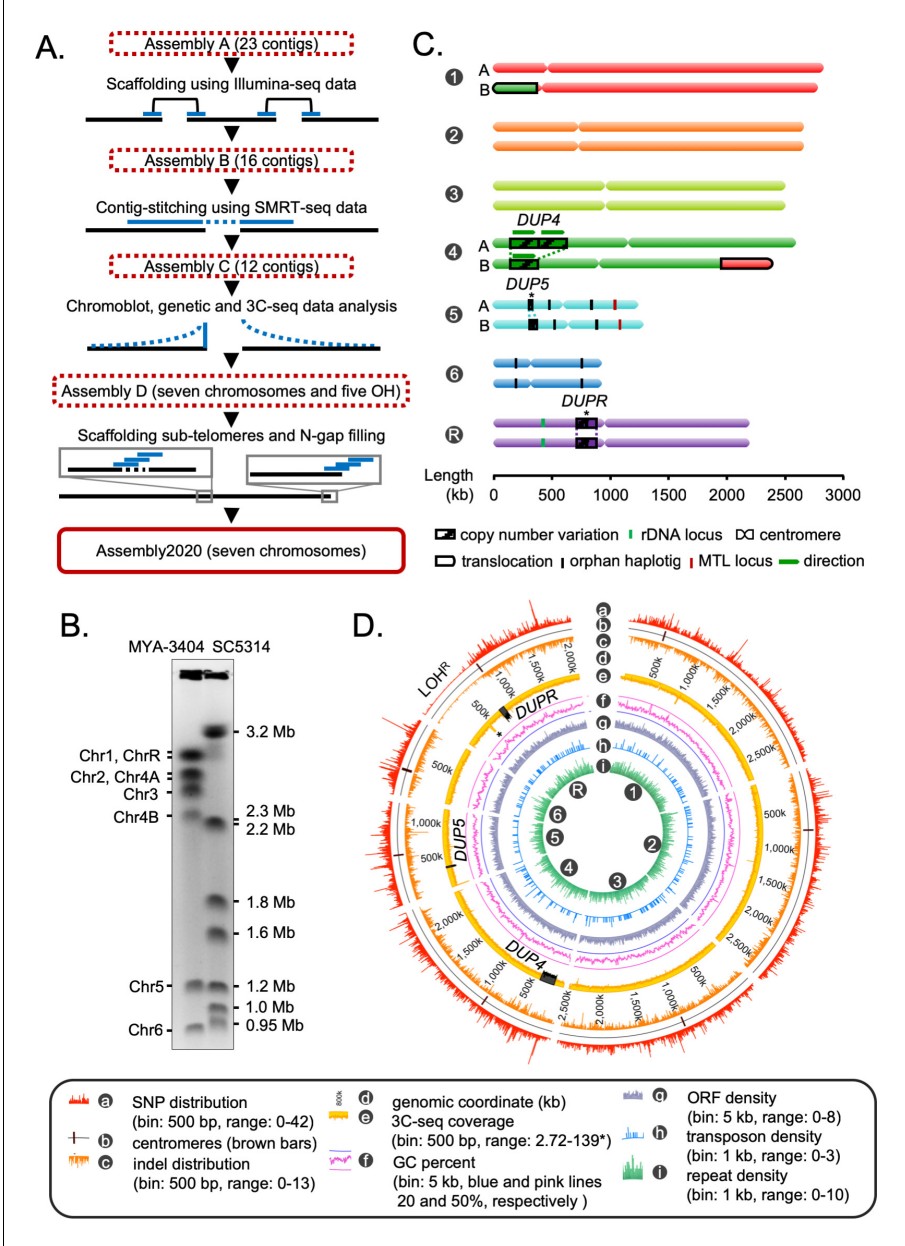

**Figure 1.** Construction of the gapless assembly of *C. tropicalis* type strain MYA-3404 in seven chromosomes. (**A**) Schematic showing the stepwise construction of the gapless chromosome-level assembly (Assembly2020) of *C. tropicalis* (also see *Figure 1—figure supplement 1* and *Figure 1—figure supplement 2*). (**B**) An ethidium bromide (EtBr)-stained CHEF gel image of separated chromosomes of the *C. tropicalis* (strain MYA-3404) and *C. albicans* (strain SC5314) (Materials and methods). *C. albicans* chromosomes are used as size markers for estimation and validation of lengths and identities of *C. tropicalis* chromosomes in the newly constructed Assembly2020. (**C**) An ideogram of seven chromosomes of *C. tropicalis* as deduced from Assembly2020 and drawn to scale. The genomic location of the three loci showing copy number variations (CNVs), *DUP4, DUP5* and *DUPR* located on Chr4, Chr5 and ChrR respectively, are marked and depicted as striped box. The CNVs for which the correct homolog-wise distribution of the duplicated copy is unknown are marked with asterisks. Homolog-specific differences for Chr1 and Chr4, occurred due to an exchange of chromosomal parts in a balanced heterozygous translocation between Chr1B and Chr4B, are highlighted with black borders (also see *Figure 1—figure supplement 4C*). (**D**) A circos plot showing the genome-wide distribution of various sequence features. Very high sequence coverage at rDNA locus is clipped for more precise representation and marked with an asterisk. The online version of this article includes the following figure supplement(s) for figure 1:

*Figure 1 continued on next page*

*Figure 1 continued*

**Figure supplement 1.** Schematic of the strategies used for construction of the gapless chromosome-level assembly of *C. tropicalis*.

**Figure supplement 2.** Orphan contigs are alleles in the diploid genome of *C. tropicalis*.

**Figure supplement 3.** Schematic outline of the strategy followed for N-gap filling and scaffolding of sub-telomeres.

**Figure supplement 4.** Identification of CNVs in the *C. tropicalis* strain MYA-3404.

**Figure supplement 5.** Chromoblot, sequence coverage analysis, and haplotyping for validation of the chromosome-level genome assembly of *C. tropicalis*.

**Figure supplement 6.** Partial conservation of a LOH block in each of the *C. albicans*, *C. tropicalis* and *C. sojae* genome.

---

3C-seq data and de novo contigs (*Figure 1—figure supplement 5B*). This translocation was validated using chromoblot analysis (*Figure 1—figure supplement 5C*) as well as Illumina, and SMRT-seq read mapping (*Figure 1—figure supplement 5D*). Thus, while chromoblot analysis suggests that the actual length of ChrR is ~2.8 Mb (*Figure 1—figure supplement 5E*), the assembled length is 2.1 Mb (*Figure 1C*). Considering the length of the rDNA locus is ~700 kb in *C. albicans* (*Jones et al., 2004*), we reason that the difference between the assembled length and actual length (derived from chromoblot analysis) of ChrR in *C. tropicalis* can be attributed to the presence of the repetitive rDNA locus of ~700 kb, which is not completely assembled in Assembly2020.

Next, we performed phasing of the diploid genome of *C. tropicalis* using SMRT-seq and 3C-seq data to identify the homolog-specific variations (Materials and methods). This analysis produced 16 nuclear contigs, which were colinear with the chromosomes of Assembly2020, except for the previously validated heterozygous translocation between Chr1 and Chr4 (*Figure 1—figure supplement 5F*). To characterize the sequence variations in the diploid genome of *C. tropicalis,* we identified the single nucleotide polymorphisms (SNPs) and insertion-deletion (indel) mutations (Materials and methods). Intriguingly, we detected a long chromosomal region depleted of SNPs and indels on the left arm of ChrR (*Figure 1D*). We named this region that lost heterozygosity on ChrR as LOH$^R$. Strikingly, we found parts of the syntenic region of LOH$^R$ to be SNP and indel depleted in the *C. sojae* strain NCYC-2607, a closely related species of *C. tropicalis*, as well as in *C. albicans* reference strain SC5314 (*Figure 1—figure supplement 6*). We also identified the genome-wide distribution of transposons and simple repeats but could not detect preferential enrichment of these sequence elements at any specific genomic location in *C. tropicalis* (*Figure 1D*). Together, we demonstrate, for the first time, multiple CNVs, a long-track LOH, and evidence of a heterozygous reciprocal translocation event in the diploid genome of *C. tropicalis.* Possible implications of these events in conferring virulence and drug resistance in this successful human fungal pathogen remain to be explored.

## Conserved principle of the spatial genome organization in *C. tropicalis* and *C. albicans*

Indirect immunofluorescence imaging of the *C. tropicalis* strain (CtKS102) expressing Protein-A tagged CENP-A$^{Cse4}$ suggested that centromeres are clustered and localized at the periphery of the DAPI-stained nuclear DNA mass as a single punctum (*Figure 2A–B*). We mapped 3C-seq data (Materials and methods), that were generated using DpnII, to the Assembly2020 to construct the genome-wide chromatin contact map of *C. tropicalis*. The resultant heatmap depicts high signal intensities along the diagonal, indicating that the intrachromosomal interactions are generally stronger than interchromosomal interactions, as observed before (*Figure 2C*; *Duan et al., 2010*). However, the most striking feature of the heatmap is the presence of conspicuous puncta in the interchromosomal areas, which signify strong spatial proximity between centromeres (*Figure 2C–D*). The aggregate signal analysis further reiterated the enrichment of centromere-centromere interactions (*Figure 2E*). Strikingly, we also noted the enrichment of telomere-telomere interactions as compared to the neighboring regions (*Figure 2C–E*). Statistical comparison was then performed between these telomere-telomere interactions and bulk chromatin, which revealed that the interchromosomal telomeric interactions were significantly greater than the all interchromosomal interactions (Mann-Whitney U test P value = $1.129 \cdot 10^{-11}$) (*Figure 2—figure supplement 1A*). On the other hand, *cis* interactions between the two telomeres of an individual chromosome (intrachromosomal

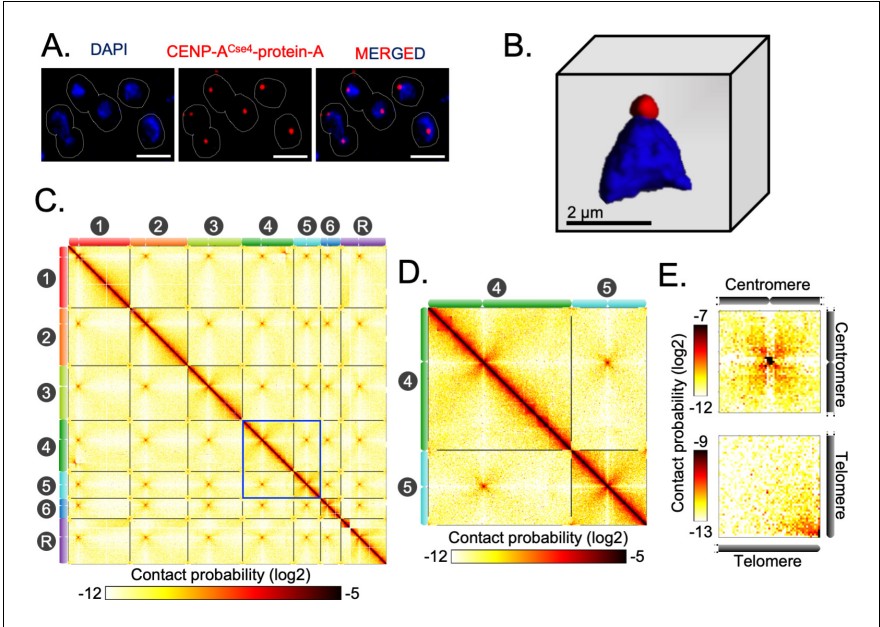

**Figure 2.** Spatial genome organization reveals centromere-centromere and telomere-telomere contacts in *C. tropicalis*. (**A**) A representative field image of *C. tropicalis* (strain CtKS102) cells expressing Protein-A tagged CENP-A$^{Cse4}$. CENP-A signals (red) were obtained using anti-Protein A antibodies by indirect immuno-fluorescence microscopy. Nuclei of the corresponding cells were stained by DAPI (blue). The images were acquired using a DeltaVision imaging system (GE) and processed using FIJI software (*Schindelin et al., 2012*). Scale, 2 μm. (**B**) A 3D reconstruction showing clustered kinetochores marked by CENP-A$^{Cse4}$ (red) at the periphery of the DAPI-stained nucleus (blue) using Imaris software (Oxford Instruments) in *C. tropicalis*. Scale, 2 μm. (**C**) A genome-wide contact probability heatmap (bin size = 10 kb) generated using 3C-seq data. Chromosome labels and their corresponding ideograms are shown on the axes of the heatmap. Colorbar represents the contact probability in the log2 scale. (**D**) Zoom in view of heatmap showing Chr4 and Chr5 from panel C (blue box). (**E**) Heatmaps plotted from aggregate signal analysis of matrices (bin size = 2 kb) surrounding centromere-centromere (top) or telomere-telomere interactions (bottom). *Top*, genomic loci containing mid-points of centromeres are aligned at the center ; *bottom*, genomic loci from 5′ or 3′ ends of chromosomes are aligned at the bottom right corner.

The online version of this article includes the following figure supplement(s) for figure 2:

**Figure supplement 1.** Analysis of 3C-seq data reveals interchromosomal and intrachromosomal telomeric contacts in *C. tropicalis* genome.

---

telomeric interactions) were also significantly enhanced compared to all intrachromosomal long-range (>100 kb) interactions (Mann-Whitney U test P value = $7.374 \cdot 10^{-11}$) (*Figure 2—figure supplement 1B*). All these lines of evidence prompted us to propose that *C. tropicalis* chromosomes adopt the Rabl-like configuration, a characteristic feature of the higher-order genome organization in yeasts (*Duan et al., 2010*; *Descorps-Declère et al., 2015*; *Burrack et al., 2016*).

Previously, microscopic and Hi-C studies revealed similar centromere clustering and strong physical interactions among centromeres in *C. albicans* (*Burrack et al., 2016*; *Sreekumar et al., 2019a*; *Sreekumar et al., 2019b*). This study now reveals that despite substantial karyotypic changes, a conserved principle of genome organization exists in two yeast species, *C. albicans* and *C. tropicalis*, with diverged centromere features.

## Centromere and telomere proximal loci are hotspots for complex translocations

Using the chromosome-level assemblies of *C. tropicalis* type strain MYA-3404 and *C. albicans* type strain SC5314 (ASM18296v3), we performed a detailed genome-wide synteny analysis employing four different approaches. We used two analytical tools, Symap (*Soderlund et al., 2011*) and Satsuma synteny (*Grabherr et al., 2010*), and a custom approach to identify the ICSBs based on the synteny of the conserved orthologs (*Figure 3A*). Next, we compared and validated the results

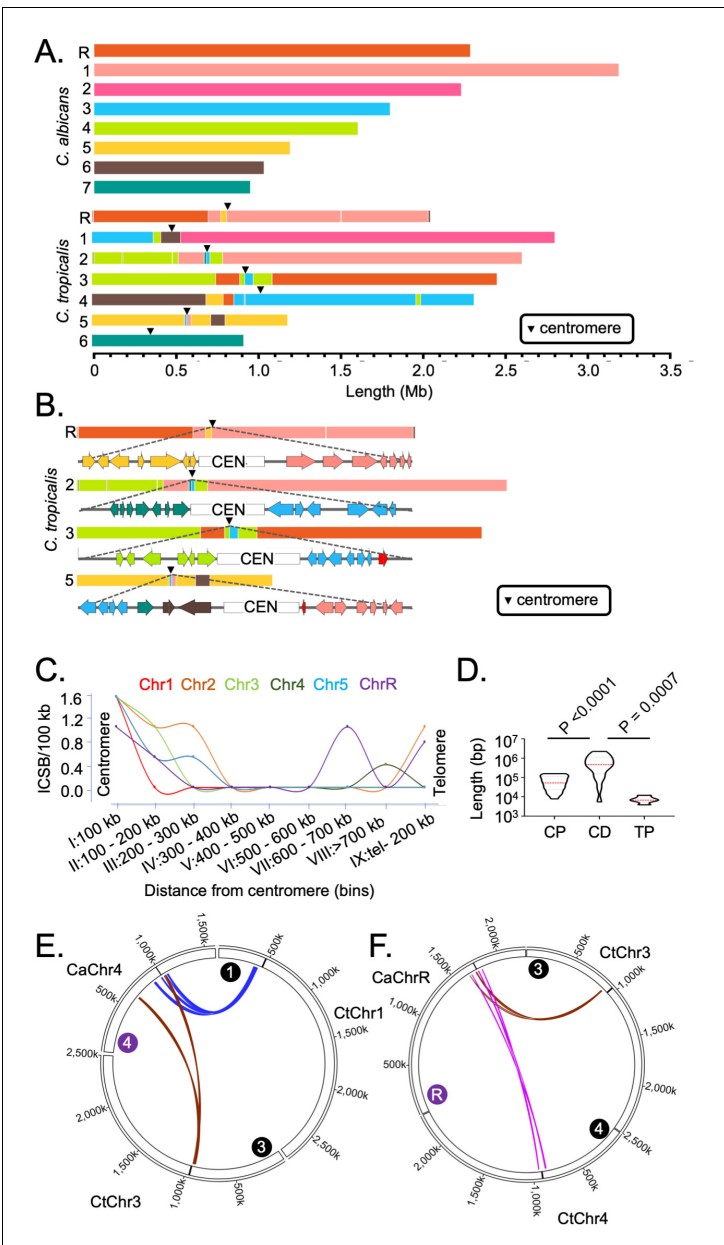

**Figure 3.** Genome-wide mapping of interchromosomal synteny breakpoints in *C. tropicalis* identifies a spatial cue for karyotype evolution. (**A**) Scaled representation of the color-coded orthoblocks (relative to *C. albicans* chromosomes) and ICSBs (white lines) in *C. tropicalis* (Materials and methods). Orthoblocks are defined as stretches of the target genome (*C. tropicalis*) carrying more than two syntenic ORFs from the same chromosome of the reference genome (*C. albicans*). The centromeres are represented with black arrowheads. (**B**) Zoom in view of the *C. tropicalis* centromere-specific ICSBs on *CEN2*, *CEN3*, *CEN5* and *CENR* showing the color-coded (relative to *C. albicans* chromosomes) ORFs flanking each centromere. *C. tropicalis*-specific unique ORFs proximal to *CEN3* and *CEN5* are shown in red. (**C**) A plot showing the chromosome-wise ICSB density, calculated as number of ICSBs per 100 kb of the C. tropicalis genome (*y*-axis), as a function of the linear distance from the centromere in nine bins. These bins are a) 0–100 kb on both sides of centromere (bin I), (b) 100–200 kb (bin II), (c) 200–300 kb (bin III), (d) 300–400 kb (bin IV), (e) 400–500 kb (bin V), (f) 500–600 kb (bin VI), (g) 600–700 kb (bin VII), (h) >700 kb to 200 kb from telomere ends (bin VIII), and i) 200 kb from the telomere ends (bin IX). Chr6 was excluded from this analysis, as it does not harbor any ICSB. (**D**) A violin plot comparing the distribution of lengths of orthoblocks (*y*-axis) at three different genomic zones: a) the centromere-proximal zone (CP), (b) the centromere-distal zone (CD), and c) telomere-proximal zone (TP). Orthoblocks, which span over more than one zone, were assigned to the zone with maximum overlap. The centromere-distal dataset was compared with the other two groups using the Mann-

*Figure 3 continued on next page*

*Figure 3 continued*

Whitney U test and the respective *P* values are mentioned. (**E - F**) Circos plots representing the convergence of centromere-proximal ORFs of *C. tropicalis* chromosomes near the centromeres (*CEN4* and *CEN7*) of *C. albicans*. Chromosomes of *C. tropicalis* and *C. albicans* are marked with black and purple filled circles at the beginning of each chromosome, respectively.

The online version of this article includes the following figure supplement(s) for figure 3:

**Figure supplement 1.** Genome-wide synteny analysis between *C. albicans* and *C. tropicalis* suggests evidence of inter-centromere translocations in the last common ancestor.

---

obtained from our custom approach of analysis with another published tool Synchro (***Drillon et al., 2014***). Considering the *C. albicans* genome as the reference, all four methods of analyses suggest that six out of seven centromeres (except *CEN6*) of *C. tropicalis* are located proximal to multiple ICSBs (***Figure 3A***, ***Figure 3—figure supplement 1A***). Although it appears that *CtCEN6* escaped inter-centromeric translocations, synteny analysis suggested that a chromosomal region carrying three consecutive *CtCEN6*-proximal ORFs was lost in the *C. albicans* genome (***Figure 3—figure supplement 1B***). Strikingly, these ICSBs are rare at the chromosomal arms (***Figure 3A***). ORF-level synteny analysis further revealed that four out of seven centromeres (*CEN2*, *CEN3*, *CEN5*, and *CENR*) in *C. tropicalis* are precisely located at the ICSBs (***Figure 3—figure supplement 1C***), while multiple ICSBs are located within ~100 kb of other two centromeres (***Figure 3A***). Additionally, a convergence of orthoblocks from as many as four different chromosomes of *C. albicans* was detected within 100 kb of *C. tropicalis* centromeres (***Figure 3B***). It is important to note that by using the *C. tropicalis* genome as the reference, all centromeres of *C. albicans*, except *CaCEN2*, were found to be associated with ICSBs (***Figure 3—figure supplement 1D***). Taken together, centromeres of both these species are found to be associated with chromosomal translocations.

To correlate the frequency of translocations with the spatial genome organization, we quantified ICSB density (the number of ICSBs per 100 kb of the genome) for different zones across the chromosome for all chromosomes except CtChr6 (***Figure 3C***). Our analysis reveals that the ICSB density is maximum at the centromere-proximal zones for all six chromosomes, but drops sharply at the chromosomal arms. However, the ICSB density near the telomere-proximal zone for Chr2, Chr4, and ChrR shows an increase compared to the chromosomal arms, albeit at a lower magnitude than centromeres. We also compared the lengths of orthoblocks across three different genomic zones - the centromere-proximal (0–300 kb from the centromere on both sides), centromere-distal (>300 kb from the centromere to 200 kb away from the telomere ends), and telomere-proximal (0–200 kb from the telomere ends) zones. This analysis further reveals that the lengths of the orthoblocks located proximal to centromeres and telomeres are significantly smaller than orthoblocks located at the centromere-/telomere-distal zones (***Figure 3D***).

We further probed into the consequences of strong inter-centromeric interactions, as described above. Synteny analysis across centromere-proximal regions of the two species hints that inter-centromeric translocations may have occurred in the common ancestor of *C. albicans* and *C. tropicalis*. If such is the case, the centromere-proximal ORFs of different chromosomes in *C. tropicalis* should have converged on the *C. albicans* genome. Indeed, we identified at least ten loci where a convergence of *C. tropicalis* ORFs from different chromosomes had taken place in *C. albicans* (***Figure 3—figure supplement 1E***). Intriguingly, we found four such loci that are proximal to the centromeres (*CEN3*, *CEN4*, *CEN7*, and *CENR*) in *C. albicans* (***Figure 3E–F***, ***Figure 3—figure supplement 1F–G***). This observation strongly supports the possibility of inter-centromeric translocation events in the common ancestor of *C. albicans* and *C. tropicalis*. Additionally, the other four centromeres in *C. albicans* are located proximal to ORFs, orthologs of which are also proximal to the centromeres in *C. tropicalis* (***Figure 3—figure supplement 1E***). We posit that the ancestral HIR-associated centromeres were lost in *C. albicans*, and ENCs formed proximal to the ancestral centromere loci on unique DNA sequences. A similar centromere type transition within two isolates of *C. parapsilosis*, another species of the CUG-Ser1 clade, has been recently reported (***Ola et al., 2020***).

## Rapid transition in the centromere type within the members of the CUG-Ser1 clade

Since multiple translocation events near centromeric regions of the *C. tropicalis* genome could be detected, we hypothesized that complex translocations between HIR-associated centromeres in the common ancestor of *C. albicans* and *C. tropicalis* led to the loss of HIR and the evolution of unique centromere types observed in *C. albicans* and *C. dubliniensis.* However, the genomic rearrangements are rare events, even at the evolutionary time scale. Therefore, if HIR-associated centromeres are to be the ancestral state from which unique centromeres were derived, some other closely related species should have retained HIR-associated centromeres. Indeed, we identified eight HIR-associated structures, in the reference genome of *C. parapsilosis* strain CDC317 (ASM18276v2). Identification of the HIR-associated structures present at the intergenic and transcription-poor regions, one each on all eight chromosomes, suggests that these loci are the putative centromeres of *C. parapsilosis.* Indeed, it was recently reported that all eight CENP-A$^{Cse4}$ enriched centromeres in the CLIB214 strain of *C. parapsilosis* are located at HIR-associated loci (*Ola et al., 2020*). Based on these lines of evidence, we conclude that the common ancestor of *C. albicans* and *C. tropicalis* possibly carried HIR-associated centromeres. Surprisingly, two centromeres in another isolate (90-137) of *C. parapsilosis* have been shown to be formed on non-HIR-associated loci (*Ola et al., 2020*). However, the driving force triggering polymorphisms in centromere locations within the same species is yet to be understood.

Although IRs are present in *CEN4*, *CEN5*, and *CENR* of *C. albicans,* these sequences are not homogenized like the HIR-associated centromeres in *C. tropicalis* (*Figure 4A*). To study the presence of HIRs in *C. sojae* (NCYC-2607), a sister species of *C. tropicalis* (*Shen et al., 2018*), we assembled its genome into 42 contigs, including seven chromosome-length contigs (Materials and methods). Using this assembly, we identified seven putative centromeres in *C. sojae* as intergenic and HIR-associated loci syntenic to the centromeres in *C. tropicalis* (*Figure 4—figure supplement 1A–C*). Each of these seven putative centromeres in *C. sojae* consists of a ~2 kb long CC region flanked by 3–12 kb long inverted repeats (*Supplementary file 5*). Using a similar approach, we identified six HIR-associated centromeres in the publicly available genome assembly (ASM332773v1) of *Candida viswanathii*, another species closely related to *C. tropicalis* (*Figure 4—figure supplement 1D–E*, *Supplementary file 6*; *Tsui et al., 2008*). A dot-plot analysis identified the presence of homologous sequences shared across IRs but not among the CC elements (*Figure 4A*) of the HIR-associated centromeres present in *C. tropicalis* and the putative centromeres of *C. sojae* and *C. viswanathii* (*Supplementary file 7*). Moreover, we detected extensive structural conservation in centromere DNA elements, especially among IRs within an individual species (*Figure 4—figure supplement 2A*). These structural feature of IRs are also significantly conserved across the three species, *C. tropicalis*, *C. sojae*, and *C. viswanathii* (*Figure 4—figure supplement 2B*).

Cloning of a full-length centromere of *C. tropicalis* in a replicative plasmid facilitated de novo CENP-A$^{Cse4}$ deposition but failed to do so when the native IRs were replaced with Ca*CEN5* IRs (*Chatterjee et al., 2016*). This result indicated DNA sequence specificity is required for centromere function in *C. tropicalis.* To identify the DNA sequence as a putative genetic element, we analyzed centromere DNA sequences of all three *Candida* species with HIR-associated centromeres and the unique centromeres of *C. albicans* for the presence of any conserved motif(s) (Materials and methods). This analysis identified a highly conserved 12-bp motif (dubbed as IR-motif) (*Figure 4B*) clustered specifically at centromeres but not anywhere else in the entire genome of *C. tropicalis, C. sojae* and *C. viswanathii* (*Figure 4C–D*, *Figure 4—figure supplement 2C*). On the contrary, the IR-motif density at centromeres in *C. albicans* remains approximately an order of magnitude lower than that of *C. tropicalis* (*Figure 4C*). This observation indicates a potential function of IR-motifs in the regulation of de novo CENP-A$^{Cse4}$ loading in *C. tropicalis.* Moreover, this *CEN*-enriched motif found at IRs is absent at central core region in *C. tropicalis* (*Figure 4E*) and at the putative centromeres in *C. sojae* and *C. viswanathii* (*Figure 4—figure supplement 2D*). Additionally, we noted that the direction of the IR-motif is diverging away from the central core in *C. tropicalis* (*Figure 4—figure supplement 2E*) as well as in the other two species (*Figure 4—figure supplement 2F*). The conserved structure and organization of the IR-motif sequences in the HIR-associated centromeres of three *Candida* species suggest an inter-species conserved function of the IR DNA sequence. However, the clusters of IR-motifs are located at a variable distance from CC in these species (*Figure 4—*

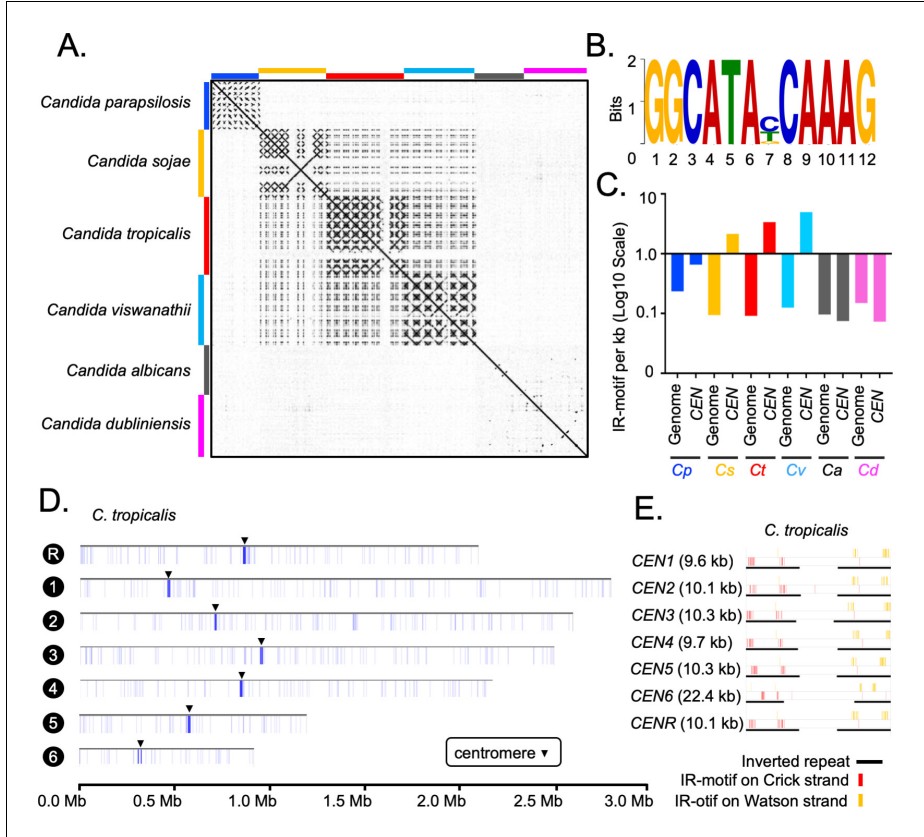

**Figure 4.** Genome-wide analysis of centromere DNA sequences across the CUG-Ser1 clade reveals the emergence of unique centromeres from an ancestral homogenized inverted repeat-associated centromere type. (A) A dot-plot matrix representing the sequence and structural homology among species of the CUG-Ser1 clade was generated using Gepard (Materials and methods). (B) A logo plot showing the 12-bp-long IR-motif, identified using MEME-suit (Materials and methods). (C) The distribution of IR-motif density on centromere DNA sequences and across the entire genome of each species was calculated as the number of motifs per kb of DNA (Materials and methods). Note that *C. albicans* and *C. dubliniensis* centromeres that form on unique and different DNA sequences do not contain the IR-motif. (D) IGV track images showing the IR-motif density across seven chromosomes of *C. tropicalis*. The location of the centromere on each chromosome is marked with a black arrowhead. (E) IGV track images showing the IR-motif distribution across seven HIR-associated centromeres of *C. tropicalis.*

The online version of this article includes the following figure supplement(s) for figure 4:

**Figure supplement 1.** Identification of HIR-associated centromeres in the CUG-Ser1 clade.

**Figure supplement 2.** Inter-species conservation of centromere DNA sequences of closely related *Candida* species.

---

*figure supplement 2G*). The importance of the sequence and the density of IR-motifs on the centromere function is yet to be determined.

## Discussion

In this study, we improved the current genome assembly of the human fungal pathogen *C. tropicalis* by employing SMRT-seq, 3C-seq, and chromoblot experiments, and present Assembly2020, the first chromosome-level gapless genome assembly of this organism. We further identified three large-scale duplication events and few small-scale CNV loci in its genome, phased the diploid genome of *C. tropicalis*, and mapped SNPs and indels. We constructed a genome-wide chromatin contact map and identified significant centromere-centromere as well as telomere-telomere spatial interactions. Comparative genome analysis between *C. albicans* and *C. tropicalis* reveals that six out of seven centromeres of *C. tropicalis* are mapped precisely at or proximal to ICSBs. Strikingly, ORFs proximal to

the centromeres of *C. tropicalis* are converged into specific regions on the *C. albicans* genome, suggesting that inter-centromeric translocations may have occurred in their common ancestor. Moreover, the presence of HIR-associated putative centromeres in *C. sojae* and *C. viswanathii*, like in *C. tropicalis*, suggests that such a centromere structure is plausibly the ancestral form in the CUG-Ser1 clade but lost both in *C. albicans* and *C. dubliniensis.* We propose that loss of such a centromere structure might have occurred during translocation events involving centromeres of homologous DNA sequences in the common ancestor, to give rise to ENCs on unique DNA sequences and facilitated speciation.

Unlike other centromeres, *CEN6* of *C. tropicalis* did not seem to undergo inter-centromeric translocations. A closer analysis revealed that three *CEN6*-associated ORFs of *C. tropicalis* are absent in the *C. albicans* genome while the other flanking ORFs remain conserved. This observation can be explained by a double-stranded DNA break at the centromere followed by the fusion of broken ends resulting in the loss of those ORFs.

The availability of the chromosome-level genome assembly and improved annotations of genomic variants and genes absent in the publicly available fragmented genome assembly of *C. tropicalis* should greatly facilitate genome-wide association studies to understand the pathobiology of this organism including the cause of antifungal drug resistance. Besides, this study sheds light on how genetic elements required for de novo centromere establishment in an ancestral species could be lost in the derived lineages to give rise to epigenetically-regulated centromeres.

*C. tropicalis* is a human pathogenic ascomycete, closely related to the well-studied model fungal pathogen *C. albicans* (*Legrand et al., 2019*). These two species diverged from their common ancestor ~39 million years ago (*Kumar et al., 2017*) and evolved with distinct karyotypes (*Chatterjee et al., 2016*), having different phenotypic traits (*Cavalheiro and Teixeira, 2018*), and ecological niches (*Pappas et al., 2018*). While *C. albicans* remains the primary cause of candidiasis worldwide, systemic ICU-acquired candidiasis is primarily (30.5–41.6%) caused by *C. tropicalis* in tropical countries including India (*Chakrabarti et al., 2015*), Pakistan (*Farooqi et al., 2013*), and Brazil (*da Costa et al., 2014*). Moreover, the occurrence of drug resistance, particularly multidrug resistance, in *C. tropicalis* is on the rise (*Chakrabarti et al., 2015*; *Xiao et al., 2015*; *Gonçalves et al., 2016*). Therefore, relatively less-studied *C. tropicalis* is emerging as a major threat for nosocomial candidemia with 29–72% broad spectrum mortality rate (*Lamoth et al., 2018*). Fluconazole resistance in *C. albicans* can be gained due to segmental aneuploidy of Chr5 containing long IRs at the centromere, by the formation of isochromosomes (*Selmecki et al., 2006*), which was also identified in Chr4 with IRs at its centromere (*Todd et al., 2019*). All seven centromeres in *C. tropicalis* are associated with long IRs with the potential to form isochromosomes.

Since the mechanism of homology search during HR is positively influenced by spatial proximity and the extent of DNA sequence homology (*Agmon et al., 2013*; *Seeber et al., 2018*), at least in the engineered model systems, it is expected that spatially clustered homologous DNA sequences undergo more translocation events than other loci. Although these factors were not shown to be involved in karyotypic rearrangements during speciation, a retrospective survey in light of spatial proximity and homology now offers a better explanation. For example, the bipolar to the tetrapolar transition of the mating type locus in the *Cryptococcus* species complex was associated with inter-centromeric recombination following pericentric inversion (*Sun et al., 2017*). Similar inter-centromeric recombination has been reported in the common ancestor of two fission yeast species, *Schizosaccharomyces cryophilus* and *Schizosaccharomyces octosporus* (*Tong et al., 2019*). These examples raise an intriguing notion that centromeres serve as sites of recombination, which may lead to centromere loss and/or the emergence of ENCs. This notion is supported by the fact that DSBs at centromeres following fusion of the acentric fragments to other chromosomes led to chromosome number reduction in *Ashbya* species (*Gordon et al., 2011*) and *Malassezia* species (*Sankaranarayanan et al., 2020*). Genomic instability at the centromere can also lead to fluconazole resistance, as in the case of isochromosome formation on Chr5 of *C. albicans* (*Selmecki et al., 2006*). Additionally, breaks at the centromeres were reported to be associated with cancers in humans (*Barra and Fachinetti, 2018*).

What would be the consequence of the spatial proximity of chromosomal regions with high DNA sequence homology in other domains of life? interchromosomal contacts between chromosome pairs have been correlated with the number of translocation events in both naturally occurring populations and experimentally induced mammalian cells (*Arsuaga et al., 2004*; *Bickmore and Teague,*

*2002*; *Branco and Pombo, 2006*; *Canela et al., 2017*; *Engreitz et al., 2012*; *Hlatky et al., 2002*; *Holley et al., 2002*; *Klein et al., 2011*; *Roukos et al., 2013*; *Zhang et al., 2012*). It has been suggested that contacts between various chromosomal territories as well as their relative positions in the nucleus influence the sites and frequency of translocation events both in flies and mammals (*Engreitz et al., 2012*; *Aten et al., 2004*; *Foster et al., 2013*; *Savage, 1998*; *Savage, 2000*; *Meaburn, 2016*). While centromeres remained clustered either throughout the cell cycle or most parts of it in many fungal species, such is not the case in metazoan cells. Nevertheless, one of the well-studied translocation events, Robertsonian translocation (RT) involving fusion between arms of two different chromosomes near a centromere, is the most frequently detected chromosomal abnormality in humans (*Therman et al., 1989*). The occurrence of RT was first reported in grasshoppers (*Robertson, 1916*) and subsequently it has been implicated in the karyotype evolution in humans (*Therman et al., 1989*), mice (*Castiglia and Capanna, 2002*; *Dumas and Britton-Davidian, 2002*), and wheat (*Friebe et al., 2005*). Moreover, RTs cause sterility in humans (*Guichaoua et al., 1990*), often linked with the heterogeneity of carcinomas (*Hermsen et al., 2005*), and implicated in genetic disorders (*Mattei et al., 1984*). Intriguingly, cytological and Hi-C based evidence (*Imakaev et al., 2012*) of spatial proximity (reviewed in *Muller et al., 2019*) among the repeat-associated centromere DNA sequences (*Kalitsis et al., 2006*) in these species supports a possibility that RTs may have been guided by spatial proximity. Similarly, chromoplexy, involving a series of translocation events among multiple chromosomes without alterations in the copy number, was identified in prostate cancers (*Zhang et al., 2013*; *Baca et al., 2013*). Although fine mapping of translocation events at the repetitive regions in human cancer cells is challenging, the growing evidence that such events are associated with the formation of micronuclei (*Crasta et al., 2012*) supports the idea that the spatial genome organization may influence chromoplexy as well (*Meaburn et al., 2007*).

The identification of HIR-associated putative centromeres in *C. parapsilosis*, *C. sojae*, and *C. viswanathii* supports the idea that the unique centromeres might have evolved from an ancestral HIR-associated centromere (*Coughlan et al., 2016*; *Figure 5A*). While HIR-associated centromeres of *C. tropicalis*, *C. sojae*, and *C. viswanathii* form on different DNA sequences, a well-conserved IR-motif was identified in this study that is present in multiple copies on the centromeric IR sequences across

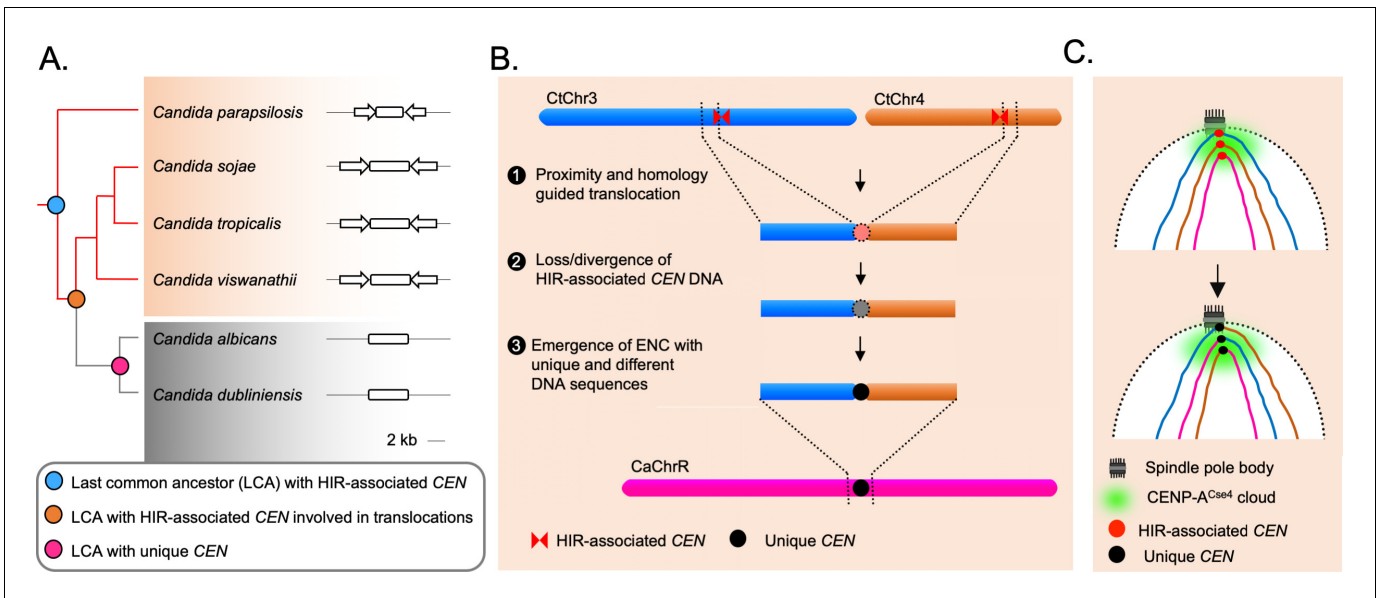

**Figure 5.** The spatial genome organization remained conserved in the CUG-Ser1 clade despite centromere type diversity. (**A**) A maximum likelihood-based phylogenetic tree of closely related CUG-Ser1 species analyzed in this study. The centromere structure of each species is shown and drawn to scale. (**B**) A model showing possible events during the loss of HIR-associated centromeres and emergence of the unique centromere type through inter-centromeric translocations possibly occurred in the common ancestor of *C. tropicalis* and *C. albicans*. The model is drawn to show translocation events involving two *C. tropicalis* chromosomes (CtChr3 and CtChr4) as representatives, which can be mapped proximal to the centromere on *C. albicans* ChrR (CaChrR) as shown in *Figure 3F*. (**C**) Rabl-like chromosomal conformation is maintained despite inter-centromeric translocations that facilitated centromere type transition.

these three species. Some centromeres in *C. albicans* carry chromosome-specific IRs but lack IR-motifs. Besides, *CaCEN5* IRs could not functionally complement the centromere function in *C. tropicalis* for the de novo CENP-A[Cse4] recruitment. This indicates a possible role of the conserved IR-motifs on species-specific centromere function (*Chatterjee et al., 2016*). Therefore, the loss of HIR-associated centromeres in *C. albicans* that are only epigenetically propagated (*Baum et al., 2006*) clearly shows how the ability of de novo establishment of kinetochore assembly in an ancestral lineage can be lost in a derived lineage. However, the mechanism through which IR-motifs may regulate centromere identity remains to be explored.

Loss of HIR-associated centromeres during inter-centromeric translocations or MIR must have been catastrophic for the cell, and the survivor was obligated to activate another centromere at an alternative locus. How is such a location determined? Artificial removal of a native centromere in *C. albicans* leads to the activation of a neocentromere (*Thakur and Sanyal, 2013*; *Ketel et al., 2009*), which then becomes part of the centromere cluster (*Burrack et al., 2016*). This evidence supports the existence of a spatial determinant, known as the CENP-A cloud or CENP-A-rich zone (*Thakur and Sanyal, 2013*; *Fukagawa and Earnshaw, 2014*), influencing the preferential formation of neocentromere at loci proximal to the native centromere (*Thakur and Sanyal, 2013*; *Scott and Sullivan, 2014*). We found that the unique and different centromeres of *C. albicans* are located proximal to the ORFs, which are also proximal to the centromeres in *C. tropicalis*. This observation indicates that the formation of the new centromeres in *C. albicans* may have been influenced by spatial proximity to the ancestral centromere cluster. However, new centromeres of *C. albicans* are formed on loci with completely unique and different DNA sequences. Similar to centromeres of *C. albicans*, centromere repositioning events may lead to the formation of ENCs, which are often associated with speciation in mammals (*Rocchi et al., 2012*; *Stanyon et al., 2008*). It was found that the location of one centromere in horse varies across individuals (*Wade et al., 2009*; *Purgato et al., 2015*). Although, there are cases where ENCs formed without genomic rearrangements, the driving force facilitating centromere relocation was proposed to be associated with chromosomal inversion and translocation in certain cases (*Schubert, 2018*). Because of these reasons, it may be logical to consider the centromeres of *C. albicans* as ENCs (*Figure 5B*). Intriguingly, even after the catastrophic chromosomal rearrangements, the ENCs in *C. albicans* remain clustered similar to *C. tropicalis* (*Figure 5C*). This observation identifies spatial clustering of centromeres as a matter of cardinal importance for the fungal genome organization.

## Materials and methods

### Media

*C. tropicalis* and *C. sojae* strains (*Supplementary file 8*) used in this study were grown in non-selective YPDU (2% dextrose, 2% peptone, 1% yeast extract, and 0.01% uracil), and incubated at 30°C at 180 rpm. For growing *C. albicans* strains, YPD media was supplemented with 0.1 mg/mL of uridine. The transformation of *C. tropicalis* was performed as described previously (*Chatterjee et al., 2016*). The selection of transformants was based on prototrophy for the metabolic markers used. In the case of selection for the antibiotic marker (*CaSAT1*), conferring nourseothricin (NTC) resistance, growth media was supplemented with 100 µg/mL NTC (NTC; Werner Bioagents, CAS No. 96736-11-7). Recycling of the *CaSAT1* marker was done by growing the NTC resistant strains in YPMU (4% maltose, 2% peptone, 1% yeast extract, and 0.01% uracil) and segregants which are NTC sensitive were selected by patching them on YPDU and YPDU supplemented with NTC. For counter selection against *CaURA3*, the 5-Fluoroorotic Acid (5-FOA; Sigma-Aldrich, CAS No. 207291-81-4) was used at 1 mg/mL concentration. The strains, primers, and plasmids used in this study are listed in *Supplementary files 8*, *9,* and *10*, respectively.

### Pulsed-field gel electrophoresis

*C. tropicalis* strain MYA-3404 and *C. albicans* strain SC5314 were grown until the exponential phase ($\sim 2 \times 10^7$ cells/mL). Cells were washed with 50 mM EDTA and counted with a hemocytometer. Approximately $6 \times 10^8$ cells were used for the preparation of 1 mL genomic DNA plugs. The plugs were made according to the instruction manual protocol (Bio-Rad, Cat No. 170–3593) with CleanCut Agarose (0.6%) and the lyticase enzyme provided by the kit. A 0.6% pulsed field certified agarose

gel was prepared using 0.5x TBE buffer (0.1 M Tris, 0.09 M Boric acid, 0.01 M EDTA, pH 8.0) and PFGE was performed on contour-clamped homogeneous electric field (CHEF) system using CHEF-DR II (Bio-Rad) module. The running conditions used were as follows: block-I at 100–200 s for 24 hr at 4.5 V/cm/120˚, block-II at 200–400 s for 48 hr at 2.5 V/cm/120˚, block-III at 600–800 s for 120 hr at 2.5 V/cm/120˚. The gel was stained with ethidium bromide (EtBr) and analyzed by Quantity One software (Bio-Rad).

## Indirect immunofluorescence microscopy

Subcellular localization of Protein-A tagged CENP-A$^{Cse4}$ with DAPI (4′,6-diamidino-2-phenylindole) stained nuclear mass was performed in *C. tropicalis* strain CtKS102 following the method described previously for *C. albicans* (*Sanyal and Carbon, 2002*). Asynchronously grown *C. tropicalis* cells were fixed with the 1/10$^{th}$ volume of formaldehyde (37%) for 1 hr at room temperature. Antibodies used were diluted as follows: 1:1000 for rabbit anti-Protein A (Sigma, Cat No. P3775). The dilutions for secondary antibodies used were Alexa flour 568 goat anti-rabbit IgG (Invitrogen, Cat No. A11011) 1:1000. Antibody dilutions were prepared in 5% skimmed milk (HiMedia, Cat No. GRM1254) solution in 1x phosphate buffered saline (PBS) pH 7.4 (137 mM NaCl, 2.7 mM KCl, 10 mM Na$_2$HPO$_4$, 1.8 mM KH$_2$PO$_4$).

## Preparation of high molecular weight genomic DNA

Briefly, 50 OD$_{600}$ equivalent (1 OD$_{600}$ = ~2 × 10$^7$ cells) cells were collected, washed with chilled 50 mM EDTA pH 8.0 and flash-frozen with liquid nitrogen. Next, the cell pellet was lyophilized. Then a volume equivalent to 5 mL of glass beads was added to the tube and vortexed till the pellet turns powdery. Then 20 mL Cetyltrimethyl ammonium bromide (CTAB) extraction buffer (100 mM Tris-HCl pH 7.5, 0.7 M NaCl, 10 mM EDTA, 1% CTAB powder, 1% 2-Mercaptoethanol) was added, and the tube was incubated at 65˚C for ~30 min with occasional mixing by inverting the tube. Subsequently the tube was chilled on ice for 10 min, and the supernatant was transferred into another tube. An equal volume of chloroform was mixed with the supernatant gently inverting for 5 to 10 min. The mix was then centrifuged at 3200 rpm for 10 min, and the aqueous phase was carefully pipetted out using cut tips to a fresh tube. An equal volume of isopropanol was added into the supernatant and mixed gently until white thread-like structures appeared. The mix was incubated at −20˚C for 1 hr and centrifuged at 3200 rpm for 10 min to pellet the DNA. The pellet was washed twice with freshly prepared 70% ethanol and air-dried. The dried pellet was dissolved in 1 mL of 1x TE containing RNase A to a final concentration of 100 µg/mL and incubated at 37˚C for 30 to 45 min. Sodium acetate solution was added into the mix to a final concentration of 0.5 M, and the solution was transferred to several 1.5 mL tubes in the aliquots of 0.4 mL each. An equal volume of isopropanol was added to each tube, mixed gently, and centrifuged at 13,000 rpm for 15 min. The supernatant was decanted, and the DNA pellet was washed with 70% ethanol. The pellet was air-dried and finally dissolved in 200 µL of 1x TE buffer. The quality of the isolated DNA was determined by performing PFGE analysis (switching time 1–25 s, at 5.8 V/cm/120˚ for 24 hr, 1% agarose gel) on CHEF-DR II module (Bio-Rad).

## Oxford Nanopore sequencing of *C. sojae* strain NCYC-2607

High molecular weight genomic DNA was isolated from yeast cells, and the average length of the DNA fragments of the genomic DNA was checked on a CHEF gel using a CHEF-DR II system (Bio-Rad). Next, the DNA sample was quantified by NanoDrop (ND-1000 Spectrophotometer, NanoDrop Technologies) and Qubit 3 fluorometer (Thermo Fisher Scientific) using dsDNA HS assay kit (Thermo Fisher Scientific, Cat No. Q33230). An appropriate amount of DNA was taken forward for library preparation as per the manufacturer's instructions using reagents included in SQK-LSK109 and EXP-NBD103/EXP-NBD104 kits. DNA samples were then pooled together on a single R9 flow-cell, and sequenced by the MinION system (Oxford Nanopore Technologies). The fragmentation step was skipped to retain the longer fragments. The raw reads were taken forward for base calling using Guppy version 3.1.5. A total of 92320 reads containing 530421800 bp were generated.

## Illumina sequencing of *C. sojae* strain NCYC-2607

DNA was quantified by Qubit 3 fluorometer (Thermo Fisher Scientific) using a dsDNA HS assay kit (Thermo Fisher Scientific, Cat No. Q33230). Approximately 100 ng of intact DNA was enzymatically fragmented by targeting 250–500 bp fragment size. The DNA fragments with overhangs resulting from fragmentation were end-filled. The 3' to 5' exonuclease activity of end-repair mix removed the 3' overhangs, and polymerase activity filled in the 5' overhangs. To the blunt-ended fragments, adenylation was performed by adding a single 'A' nucleotide to the 3' ends. To the adenylated fragments, loop adapters were ligated and cleaved with uracil-specific excision reagent enzyme. The sample was further purified using AMPure XP beads (Beckman Coulter, Cat No. A63880), and DNA was then enriched by PCR with six cycles using NEBNext Ultra II Q5 master mix (NEB, Cat No. M0544S), Illumina universal primers, and sample-specific indexed Illumina primers. The amplified products were cleaned up by using AMPure XP beads, and the final DNA library was eluted in 15 µL of 0.1x TE buffer. One µL of the library was used to quantify the DNA concentration by Qubit 3 fluorometer using the dsDNA HS reagent. The fragment analysis was performed on Agilent 2100 Bioanalyzer (Agilent, Model G2939B), by loading 1 µL of the library into Agilent DNA 7500 chip. In this experiment, we generated 3501768 paired-end reads of 2 × 301 bp length.

## De novo genome assembly of *C. sojae* strain NCYC-2607

A total of 92320 reads containing 530421800 bp were used for the construction of a de novo assembly using Canu (*Koren et al., 2017*). Canu was run using default parameters in the trimming and the correction mode with '-genomeSize < 15 m>', which produced the genome assembly of *C. sojae* in 42 contigs. Next, to rectify the base-pair level errors, we performed five rounds of polishing of the contigs using Illumina reads with Pilon (*Walker et al., 2014*).

## SMRT sequencing on PacBio sequel system

The genomic DNA fragments of ~20 kb length were size-selected and taken forward for library preparation using SMRTbell Template Prep Kit (Part No. 100-259-100). PacBio sequencing of the *C. tropicalis* MYA-3404 genome was performed by Sequel SMRT Cell 1M (Part No. 101-008-000) using Sequel Binding Kit 2.0 (Part no. 100-862-200) and SMRT Link version 5.0.1.9585. This run generated 996041 reads with an average read length of 5.8 kb.

## Construction of assembly B

Gepard (*Krumsiek et al., 2007*) was used to generate dot matrix plots and identify areas of overlap between supercontigs. Supercontigs whose ends overlapped were identified and their sequences merged. Whole genome Illumina sequencing data of the *C. tropicalis* strain MYA-3404 were used to verify these predictions. We submitted the reads to NCBI under the BioProject accession number PRJNA604451.

## Construction of the de novo SMRT assembly and contig stitching using SMIS

The de novo SMRT assembly using 996041 PacBio raw reads was generated using Canu 1.6 (94). The program was run in the trimming and correction mode with the '-pacbio-raw<input.fastq>' option that produced 135 contigs. For stitching the contigs from Assembly B using the PacBio raw reads, we used Single Molecular Integrative Scaffolding (*Ning, 2014*) with the default options, to get a 12-contig assembly (Assembly C). Details of the assemblies produced by Canu and SMIS are presented in *Supplementary file 3*.

## Filling N-gaps

The de novo SMRT contigs were used to fill the existing N-gaps in Assembly A. We used 500 bases upstream, and downstream regions of the N-gaps as queries against a custom BLAST (*Altschul et al., 1990*) database generated using Geneious software from the de novo assembled contigs and filled these N-gaps upon the mapping of upstream and downstream query sequences on the same contig with 100% coverage and more than 95% identity. Using this approach, we filled 78 out of 104 gaps leaving 26 gaps on seven chromosomes (*Supplementary file 2*, *Figure 1—figure supplement 3A*). We suspected that the remaining gaps were repetitive regions in the genome as

immediate flanking regions identified multiple hits. To avoid this, we used a second strategy in which we used a 1 kb query sequence from either 10 kb upstream or downstream region of the N-gap, and performed a BLAST analysis against the de novo contigs generated using FALCON (*Chin et al., 2016*). All the remaining 26 gaps could be filled using this strategy (*Supplementary file 2*, *Figure 1—figure supplement 3B*). Further, to validate our claim, we confirmed the mapping of the Illumina and PacBio reads over the newly filled sequence.

## Assembly of sub-telomeric regions

To assemble the sub-telomere regions, we performed a BLAST search using the terminal 5000 bp sequence of each chromosome as queries against the de novo SMRT contigs and identified the contigs containing the 23 bp telomeric repeats specific for *C. tropicalis* (5′-TGATCGTGACATCCTTA-CACCAA-3′) as reported previously (*Butler et al., 2009*). Schematic of the sub-telomere scaffolding has been shown (*Figure 1—figure supplement 3C*).

## Mapping of the orphan haplotigs using the de novo SMRT assembly

Canu is a diploid-aware genome assembler (*Koren et al., 2017*), which generates two contigs from a heterozygous locus. Therefore, we used the Canu generated contigs (SMRT assembly) to map the orphan haplotigs as heterozygous regions of the genome (see *Figure 1—figure supplement 1H*). Heterozygosity of the orphan haplotigs was demonstrated by the Illumina read coverage (*Figure 1—figure supplement 2B*). For this analysis, the 3C-seq reads were mapped on the OHs and a control locus of Chr1 using Bowtie2 (*Langmead and Salzberg, 2012*). The number of mapped reads were counted using the bamCoverage utility from deepTools2 (*Ramírez et al., 2016*) and plotted using boxplotR (*Spitzer et al., 2014*).

## Pilon polishing of the genome assembly

The final telomere-to-telomere assembled chromosomes were polished through Pilon (*Walker et al., 2014*) using the Illumina reads obtained from the 3C-seq experiment. Pilon corrected base-pair level assembly errors and validated 99.5–99.8% bases of the seven chromosomes. The polishing step was repeated six times when the improvement stalled.

## Construction of aneuploids for confirmation of heterozygosity of the OHs

We constructed *C. tropicalis* strains monosomic for Chr5 and used them to demonstrate that loss of one homolog of Chr5 leads to loss of one of the two alleles of the orphan contigs: contig14 and contig16, that are mapped on Chr5. Since the *sch9* mutants in *C. albicans* were viable but lost chromosomes at a significantly higher rate than the wild-type (*Varshney et al., 2015*), we adopted the same strategy to delete both copies of *SCH9* homologs in *C. tropicalis*. Next, a reporter strain was created in this *sch9* mutant strain background of *C. tropicalis* to assay for loss of a Chr5 homolog. These strains (2 n-1) that lacked one homolog of Chr5 were used to confirm the presence of heterozygosity of orphan haplotigs (OHs) of CtChr5.

### A. Deletion of *SCH9* in *C. tropicalis*

The *SCH9* homolog in *C. tropicalis* was identified in a BLAST search using *CaSCH9* as the query sequence against the *C. tropicalis* proteome. A putative homolog of *SCH9* was located on Chr1:1994521–1996662 and encoded by the Crick strand. A deletion cassette (pKG1) for double homologous recombination-mediated deletion of *SCH9* ORF was constructed by cloning upstream and downstream homology regions in pSFS2a plasmid (*Reuss et al., 2004*). This construct was transformed into CtKS102 for the deletion of both copies of *SCH9* ORF by recycling the *CaSAT1* marker after the deletion of the first copy of *SCH9* gene. Independent colonies of the *sch9/sch9* null mutant strain (CtKG001) were confirmed using Southern hybridization (*Figure 1—figure supplement 2C–D*). Primers used in this study are mentioned in *Supplementary file 9*.

## B. Construction of a reporter strain for construction of strains with Chr5 monosomy by integration of *URA3* on Chr5

Upstream and downstream homology regions of the target intergenic locus (Chr5_497_kb) in Chr5 were amplified, and cloned into pBSCaURA3 plasmid (*Chatterjee et al., 2016*) to construct pKG2 (*Supplementary file 10*). This cassette was released by restriction digestion with BamHI and ApaI and transformed into the *sch9* mutant strain CtKG001 to construct the reporter strain CtKG002. Similarly, we integrated *CaURA3* into the target intergenic locus (Chr5_497_kb) of CtKS102 to create a control strain CtKG003. In both the strains (CtKG002 and CtKG003) the short arm (5' end) of one of the two homologs is marked with *CaURA3* marker and the long arm (3' end) carries the heterozygous *MTL* locus (*MTL*a or *MTLα*) with two distinct alleles present on two homologs. Concomitant loss of one of the *MTL* alleles together with *CaURA3* marker would indicate loss of one homolog of Chr5.

## C. Isolation and confirmation of the 2 n-1 aneuploids for Chr5

Different cell numbers ($10^5$, $10^4$, $10^3$, and $10^2$) of the reporter strain (CtKG002) and the wild-type control strain (CtKG003) were plated on complete media (CM) + 5-FOA and incubated for 48–72 hr at 30°C. Multiple FOA$^R$ colonies appeared for CtKG002 strain but no colonies appeared for the control strain CtKG003. The colonies were then patched on YPDU and CM-URA plates for confirmation of the loss of the *CaURA3* marker. Next, PCR was performed to confirm the loss of one of the *MTL* loci (*MTL*a or *MTLα)* in these colonies using a multiplex PCR strategy described previously (*Figure 1—figure supplement 2G*; *Porman et al., 2011*).

## Library preparation and sequencing of the library DNA for chromosome conformation capture (3C-seq)

Wild-type *C. tropicalis* strain MYA-3404 was cultured in non-selective YPDU media and 500 OD$_{600}$ equivalent cells were harvested for crosslinking. The cells were cross-linked with formaldehyde to a final concentration of 1.5% for 10 min and the cross-linking reaction was quenched by adding glycine to a final concentration of 400 mM. The crosslinked cells were centrifuged and the cell pellet was stored at −80°C till further use.

For making the 3C library of *C. tropicalis*, the cross-linked cell pellet was first resuspended in 5 mL of ice-cold 1x NEBuffer DpnII (50 mM Bis-Tris-HCl, 100 mM NaCl, 10 mM MgCl2, 1 mM DTT; pH 6 @ 25°C) and then lysed by liquid nitrogen grinding in a chilled mortar using a pestle to a fine powder. The powdered sample was scraped using a spatula into a pre-chilled tube and resuspended in 15 mL cold 1x NEBuffer DpnII. Cell lysate containing ~3 × $10^8$ cells (4 mL) was processed for 3C library preparation. This lysate was centrifuged and the pellet was resuspended in 1.5 mL of cold 1x NEBuffer DpnII and then aliquoted equally into four 1.5 mL microcentrifuge tubes. Next, the chromatin was solubilized by adding SDS to a final concentration of 0.1% in each microcentrifuge tube and the sample was incubated at 65°C for exactly 10 min. The reaction was quenched by adding 45 µL of 10% Triton X-100 per tube with gentle mixing. Chromatin was then digested with 750 units of DpnII (NEB, Cat No. R0543M; 50,000 units/mL) per tube and incubated at 37°C overnight with gentle agitation (300 rpm) on a heating block. Next day, the restriction enzyme was heat-inactivated at 65°C for 20 min. The digested chromatin fraction in each tube was ligated with 50 U of T4 DNA ligase (Invitrogen Cat No.15224090; 1 U/µL) at 16°C for 6 hr in a diluted condition (reaction volume 8 mL) to favor intra-molecular ligation of cross-linked restriction fragments. Reverse cross-linking was performed by adding 100 µL of 10 mg/mL Proteinase K (Invitrogen Cat No.25530031) per tube and incubating at 65°C overnight. Next, DNA, which constitutes the 3C library, was purified using conventional phenol-chloroform extraction and concentrated using Amicon Ultra-0.5 mL 30K centrifgal filters. About 1 µg of 3C library was used for size selection using Agencourt AMPure XP beads (Beckman Coulter) to select DNA fragments of 500–700 bp in length. The paired-end NGS library was prepared using NEBNext Ultra II kit, and sequencing was carried out using the Illumina HiSeq 2500 2 × 101 bp platform by a third party service provider.

## 3C-seq data analysis

FASTQ files containing ~75 million 2 × 101 bp paired-end 3C-seq reads were initially processed using hiclib package (http://mirnylab.bitbucket.org/hiclib/) (*Imakaev et al., 2012*). The resultant

genome-wide chromatin interaction matrix was converted to a contact probability matrix. Codes associated with the downstream analysis could be found at the Github repository (https://github.com/Yao-Chen/candida-tropicalis-analysis); copy archived at https://github.com/elifesciences-publications/candida-tropicalis-analysis; *Chen, 2020*).

## A. Mapping of reads and generation of the contact probability matrix

First, two sides of paired-end reads were separated and iteratively aligned to Assembly2020 using Bowtie2 (*Langmead and Salzberg, 2012*) with the '--very-sensitive' option. The iteration started from the first 20 bases of each read and continued with an increment of 5 bases in the subsequent iteration. Next, the aligned read pairs whose both sides had MAPQ score ≥1 were processed through the fragment filter, where self-circles, dangling ends, extra-dangling ends (maximum molecular length = 500), error pairs and PCR duplicates were excluded from downstream analysis. A genome-wide interaction matrix was generated using the remaining unique valid pairs (bin size = 2–10 kb). The bin filter removed bins with <50% sequence information in the reference genome and 1% bins with low read coverage. The matrix was iteratively corrected for biases and eventually converted to a contact probability matrix where the sum of each row/column approximates 1.

## B. Aggregate signal analysis

Sub-matrices for all combinations of centromere-centromere interactions across different chromosomes were extracted from the genome-wide contact probability matrix. Genomic loci containing mid-points of centromeres were first aligned and then all the sub-matrices were stacked on top of each other and averaged. Similar analysis was performed for all telomere-telomere interactions (both intra and interchromosomal telomeric interactions) where sub-matrices for telomere-telomere interactions across all chromosomes were extracted, stacked and averaged.

## C. Analysis of telomeric interactions

To investigate interchromosomal telomeric interactions, a histogram of all interchromosomal interactions (excluding zero values) were generated (bulk chromatin). Mean contact probabilities of all interchromosomal interactions as well as all interchromosomal telomeric interactions were computed, where 5' and 3' end bins on each chromosome were taken as telomeric bins. Similarly, a histogram of all intrachromosomal long-range interactions (excluding zero values) was generated (bulk chromatin), where long-range interactions were defined as interactions between loci separated at a distance of >100 kb. Intrachromosomal telomeric interactions were taken as interactions between two loci that were close to two telomeres of an individual chromosome respectively (sum of distances between each locus to the nearest telomere is ≤10 kb). Mann-Whitney U test was used to compare interchromosomal or intrachromosomal telomeric interactions to the bulk chromatin.

## D. Contig scaffolding

3C-seq reads were aligned to contig sequences and contact probability matrix was generated as described above. The 3C profile of a bin was plotted using values in a single row from the contact probability matrix. It is well-established that contact frequency generally shows a distance-dependent decay (*Dekker et al., 2002*). Therefore, the connectivity between two contigs can be inferred by investigating the contact probabilities between the terminal bin of a contig and loci on the other contig.

## Identification of SNPs, indels and CNVs

### A. Detection of SNPs and indels

The SNPs and indels were identified using GATK software (*McKenna et al., 2010*) with the paired-end Illumina reads obtained from the 3C-seq experiment in a 12 cores Ubuntu 16.4 system with 96 GB memory. Briefly, the 3C-seq reads were mapped to Assembly2020 using Bwa-mem (*Li, 2013*) paired-end alignment mode following sorting of the resulting SAM file with Picard (https://broadinstitute.github.io/picard/), SAM to BAM conversion using SAMtools (*Li et al., 2009*), and duplicate marking using 'MarkDuplicates' utility of Picard. Next, we used GenomeAnalysisTK.jar (version 3.8.0) to call the variants with '-ploidy 2' option, SNPs were extracted, filtered with '--filterExpression 'QD <2.0 || FS >60.0 || MQ <40.0 || MQRankSum <−12.5 || ReadPosRankSum <−8.0 || SOR > 4.0' −

`filterName` 'basic_snp_filter'' option following base quality score recalibration. Similarly indels were extracted, filtered with '–`filterExpression` 'QD <2.0 || FS >200.0 || ReadPosRankSum <−20.0 || SOR > 10.0' –`filterName` 'basic_indel_filter'' option following base quality score recalibration. The data tracks were visualized using IGV (*Robinson et al., 2011*) and presented using Circa software.

## B. Read coverage plot and CNV detection

To generate a genome-wide read coverage plot, the 3C-seq reads were mapped to Assembly2020 using Bowtie2 (*Langmead and Salzberg, 2012*) paired-end alignment mode with '–`end-to-end`' and '–`very-sensitive`' option. The resultant SAM file was converted to BAM format and sorted using SAMtools (*Li et al., 2009*). Next, the mapped reads were counted using deepTools2 (*Ramírez et al., 2016*) bamCoverage utility with the BPM normalization method, and the resulting BED file was used for downstream calculations or visualization in IGV (*Robinson et al., 2011*). To detect CNVs throughout the *C. tropicalis* genome, the sorted BAM file, generated from coverage analysis above, was processed by 'CNAtraLite' option of CNAtra tool (https://github.com/AISKhalil/CNAtra) (*Khalil et al., 2020*). where the MAPQ filter was disabled. The read depth signal (bin size = 1 kb) and the estimated copy numbers were then plotted for each chromosome by 'CNVsTrackPlot' function of 'CNAtraLite'. In this analysis, regions whose estimated copy numbers are <1.5 or>2.5 are considered as CNVs.

## Haplotype analysis

The FALCON, FALCON-Unzip (*Chin et al., 2016*), and FALCON-Phase (*Kronenberg, 2018*) from the pb-assembly suite were run locally in a 12 core Ubuntu 16.4 system with 96 GB memory according to the instruction provided (https://github.com/PacificBiosciences/pb-assembly) (*Chin et al., 2016*; *Wenger et al., 2019*) The configuration files used for running FALCON, FALCON-Unzip and FALCON-Phase will be available upon request. Briefly, FALCON was run using modified fcrun.cfg with the input option 'pa_DBdust_option = true', and 'pa_fasta_filter_option = streamed-internal-median'. Next, data partitioning was performed with 'pa_DBsplit_option=-x500 -s100' and 'ovlp_DBsplit_option=-x500 -s100', repeat masking was performed using 'pa_HPCTANmask_option = -k18 -h480 -w8 -e.8 -s100', 'pa_HPCREPmask_option = -k18 -h480 -w8 -e.8 -s100', and 'pa_REPmask_code = 0,300;0,300;0,300' options. Preassembly was generated using the following parameters: 'genome_size = 15000000', 'seed_coverage = 20', 'length_cutoff = 100', 'pa_HPCdaligner_option=-v -B128 -M24', 'pa_daligner_option=-e.8 -l1000 -k18 -h480 -w8 -s100 -T10', 'falcon_sense_option=–`output-multi` –`min-idt` 0.70 –`min-cov` 2 –`max-n-read` 1800', 'falcon_sense_greedy = False'. Next, Pread overlapping was performed using 'ovlp_daligner_option=-e.96 -l1000 -k24 -h1024 -w6 -s100', and 'ovlp_HPCdaligner_option=-v -B128 -M24'. Next, the final assembly was generated using 'overlap_filtering_setting=–`max-diff` 100 –`max-cov` 100 –`min-cov` 2' and 'length_cutoff_pr = 500'. Next, phasing of haplotypes was performed using FALCON-Unzip and FALCON-Phase as described (https://github.com/PacificBiosciences/pb-assembly) (*Chin et al., 2016*; *Wenger et al., 2019*).

## Assessment of the genome assembly completeness using BUSCO

BUSCO (*Simão et al., 2015*) version 3.0.2 was run against ascomycota_odb9 database using the following script: python./scripts/run_BUSCO.py -i genome.fasta -o BUSCO_output -l/Path_to_llineage_dir/ -m genome -c 1 -sp candida_tropicalis.

## Synteny analysis

Genome-wide synteny analysis was performed using Symap (*Soderlund et al., 2011*) with the parameters as (a) Min. dots 3 (minimum number of anchors required to define a synteny block), (b) top N 2 (retain the top N hits for every sequence region, as well as all hits with score at least 80% of the Nth), (c) BLAT args: '-minScore = 30 -minIdentity = 70 -tileSize = 10 -qMask = lower -maxIntron = 10000'. The Satsuma synteny and Synchro software were run using default parameters. For the custom approach to map the interchromosomal synteny breakpoints (ICSBs), first, the single-copy orthologs were identified using OrthoFinder (*Emms and Kelly, 2015*), then the corresponding genomic coordinates of the ortholog pairs were sorted and the ICSBs were identified. For

comparing the FALCON generated contigs with the Assembly2020 chromosomes, the dot-plot between the two assemblies was generated using the default options of Symap version 4.2.

## Identification of the putative centromeres in the members of CUG-Ser1 clade

The putative centromeres of *C. sojae* and *C. viswanathii* were identified as HIR-associated intergenic regions syntenic to centromeres of *C. tropicalis* centromeres. Briefly, the genomic loci in *C. sojae* and *C. viswanathii,* which are syntenic to the centromeres of *C. tropicalis* were scanned for the presence of inverted repeats falling in ORF-free regions using YASS (*Noé and Kucherov, 2005*) with the default parameters. Pair-wise alignments between seven random genomic loci of ~10 kb length, LR, CC, or RR DNA elements were performed using Clustal Omega (*Sievers and Higgins, 2014*). Synteny dot-plot analysis for centromere DNA sequences including the flanking ORF-free region in *C. albicans, C. dublininensis* and the HIR sequences of *C. tropicalis, C. sojae, C. viswanathii,* and *C. parapsilosis* was generated using Gepard (*Krumsiek et al., 2007*) by running it in the simple mode with default parameters. The IR sequences from centromeres of *C. tropicalis* and the putative centromeres of *C. sojae* and *C. viswanathii* were analysed to identify the presence of conserved motifs using motif discovery tool MEME following the default parameters with 'ZOOPS: zero or one site per sequence' as the motif site distribution algorithm, and maximum motif width set to 12 bp. Next, we scanned for the presence of IR-motifs across the chromosomes including centromere DNA and flanking ORF-free regions in *C. albicans, C. dubliniensis,* and putative centromeres of *C. parapsilosis* using FIMO with default parameters (*Bailey et al., 2009*).

## Construction of the phylogenetic tree

The publicly available genomes and the protein fasta files (when available) of *C. albicans* (ASM254v2), *C. dubliniensis* (ASM2694v1), *C. viswanathii* (ASM332773v1), and *C. parapsilosis* (ASM18276v2) were downloaded from NCBI database. The protein fasta files for *C. tropicalis* and *C. sojae* were generated using Augustus *ab initio* protein prediction software and the python script getAnnoFasta.py (*Stanke and Morgenstern, 2005*). Because of the partially diploid nature of *C. viswanathii* genome assembly, the duplicated contigs, that carried >100 kb of DNA sequence on another contig, were identified from dot-plot analysis (self) using Symap (*Soderlund et al., 2011*), and excluded from analysis. The protein fasta files were then used as input files for running Ortho-Finder V2.3.1 (112). OrthoFinder was run using the default parameters except the -M msa option for the construction of maximum-likelihood trees using MAFFT (*Katoh et al., 2002*) and FastTree (*Price et al., 2010*). The tree topology was visualized using Evolview (*Subramanian et al., 2019*).

## Data access

All sequencing data of *C. tropicalis* and *C. sojae* reported in this study have been submitted to NCBI under the BioProject accession numbers PRJNA596050 and PRJNA604451. The sequences of seven chromosomes of *C. tropicalis* in Assembly2020 (GCA_013177555.1 ASM1317755v1) are available through GenBank accession numbers CP047869-CP047875. The contig sequences of *C. sojae* genome assembly (GCA_013177575.1 ASM1317757v1) are available through GenBank accession number WWPN00000000.

## Acknowledgements

We thank all the members of KS laboratory and AS laboratory for stimulating discussions and critical reading of the manuscript. We acknowledge S Sun and J Heitman for helping with SMRT-seq of *C. tropicalis* at the PacBio sequencing facility at Duke University. We also thank AIS Khalil for helping with CNAtra software for CNV analysis. Illumina sequencing experiments for the *C. sojae* genome were performed at Clevergene Biocorp, Bangalore, India. We also thank B Suma for confocal microscopy, JNCASR. KG acknowledges Shyama Prasad Mukherjee Fellowship from Council of Scientific and Industrial Research (CSIR), Govt. of India [07/733 (0181)/2013-EMR-I] and financial assistance from JNCASR. This project is supported by a grant (BT/PR27490/Med/29/1323/2018) from the Department of Biotechnology (DBT), Govt. of India to KS. KS acknowledges TATA innovation fellowship (BT/HRD/35/01/03/2017) and Department of Biotechnology grant in Life Science Research, Education and Training at JNCASR (BT/INF/22/SP27679/2018). Intramural funding from JNCASR is

acknowledged. This work is also supported by Nanyang Technological University's Nanyang Assistant Professorship grant and Singapore Ministry of Education Academic Research Fund Tier 1 grant [RG39/18] to AS.

## Additional information

### Funding

| Funder | Grant reference number | Author |
|---|---|---|
| Council of Scientific and Industrial Research | Shyama Prasad Mukherjee Fellowship 07/733(0181)/ 2013-EMR-I | Krishnendu Guin |
| Department of Biotechnology, Ministry of Science and Technology | BT/PR27490/Med/29/1323/ 2018 | Kaustuv Sanyal |
| Ministry of Education - Singapore | RG39/18 | Amartya Sanyal |
| Department of Biotechnology, Ministry of Science and Technology | BT/INF/22/SP27679/2018 | Kaustuv Sanyal |
| Department of Biotechnology , Ministry of Science and Technology | BT/HRD/35/01/03/2017 | Kaustuv Sanyal |
| Nanyang Technological University | Nanyang Assistant Professorship grant | Amartya Sanyal |

The funders had no role in study design, data collection and interpretation, or the decision to submit the work for publication.

### Author contributions

Krishnendu Guin, Conceptualization, Formal analysis, Writing - original draft, N-gap filling, Pilon polishing, haplotype analysis, genome-wide synteny analyses, identification of the putative HIR-associated centromeres and motif analysis, chromoblot analysis, Southern blotting, and subcellular localization experiments, Oxford Nanopore library preparation for C. sojae and generated its genome assembly.; Yao Chen, Conceptualization, Formal analysis, Validation, Writing - original draft; Radha Mishra, Constructed aneuploid strains; Siti Rawaidah BM Muzaki, Performed the 3C-seq library preparation; Bhagya C Thimmappa, Performed SMIS and Canu run and scaffolding of telomeres; Caoimhe E O'Brien, Geraldine Butler, Performed scaffolding of C. tropicalis genome in 16 contigs (Assembly B); Amartya Sanyal, Kaustuv Sanyal, Conceptualization, Supervision, Writing - original draft, Writing - review and editing

### Author ORCIDs

Krishnendu Guin  https://orcid.org/0000-0001-6957-465X
Yao Chen  https://orcid.org/0000-0003-2708-8674
Radha Mishra  https://orcid.org/0000-0001-7111-445X
Geraldine Butler  https://orcid.org/0000-0002-1770-5301
Amartya Sanyal  https://orcid.org/0000-0002-2109-4478
Kaustuv Sanyal  https://orcid.org/0000-0002-6611-4073

### Decision letter and Author response

Decision letter https://doi.org/10.7554/eLife.58556.sa1
Author response https://doi.org/10.7554/eLife.58556.sa2

# Additional files

## Supplementary files

- Source data 1. Source_data_combined.
- Supplementary file 1. Assembly C with 12 contigs.
- Supplementary file 2. Assembly of sub-telomeres and filling up N-gaps in the genome assembly of *C. tropicalis* using *de* contigs.
- Supplementary file 3. Statistics for different versions of genome the assembly of *C. tropicalis* (MYA-3404) generated in this study.
- Supplementary file 4. A comparative analysis of Assembly A and the improved Assembly2020 of *C. tropicalis*.
- Supplementary file 5. Features of centromere DNA elements in *C. sojae*.
- Supplementary file 6. Features of centromere DNA elements in *C. viswanathii*.
- Supplementary file 7. Centromere coordinates used for identifying conserved DNA sequence motifs in *Candida* species.
- Supplementary file 8. List of strains used in this study.
- Supplementary file 9. List of primers used in this study.
- Supplementary file 10. List of plasmids used in this study.
- Transparent reporting form

## Data availability

All sequencing data reported in the study and the genome assembly of C. tropicalis and C. sojae have been submitted to NCBI under the BioProject accession numbers PRJNA596050 and PRJNA604451.

The following datasets were generated:

| Author(s) | Year | Dataset title | Dataset URL | Database and Identifier |
|---|---|---|---|---|
| Guin K, Chen Y, Mishra R, Muzaki SRBM, Thimmappa BC, O'Brien CE, Butler G, Sanyal A, Sanyal K | 2020 | Candida tropicalis and Candida sojae | https://www.ncbi.nlm.nih.gov/bioproject/?term=PRJNA596050 | NCBI BioProject, PRJNA596050 |
| Guin K, Chen Y, Mishra R, Muzaki SRBM, Thimmappa BC, O'Brien CE, Butler G, Sanyal A, Sanyal K | 2020 | Whole genome sequencing of Candida tropicalis isolates | https://www.ncbi.nlm.nih.gov/bioproject/?term=PRJNA604451 | NCBI BioProject, PRJNA604451 |

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
