## [Decision Letter]

[Editors' note: this paper was reviewed by Review Commons.]

**Acceptance summary:**

A high-accuracy gapless assembly of *Candida tropicalis* is presented and compared to other candida species, revealing evolutionary changes to centromeres. The authors present evidence for interchromosomal rearrangement events near centromeres and telomeres that could have contributed to centromere evolution. 3C data shows that, as in other fungi, centromeres and telomeres cluster in 3D space, providing a possible mechanisms by which rearrangements could be favored to occur between centromeres and between telomeres.

---

## [Author Response]

Reviewer #1 (Evidence, reproducibility and clarity (Required)):The authors have tried to address sudden evolutionary jump in centromere formation of two closely related species i.e. Candida tropicalis (Ct) and Candida albicans (Ca). While Ct has homogenized inverted repeat (HIR) associated centromeres, Ca evolved to form centromeres on unique DNA elements. To address this, the authors first generated chromosome level genome assembly (Assembly2020) of Ct using whole genome Illumina sequencing, 3C sequencing, PacBio SMRT sequencing, chromoblot analysis, and genetic analysis of engineered aneuploid strains to improve previously existing fragmented genome assembly. Interestingly, even though Ca and Ct have different mechanisms to form centromeres, there spatial organization of colocalizing centromeres at the nucleus periphery remains conserved. This is also demonstrated by 3C data which shows significantly higher centromerecentromere contact probability. Authors speculate that spatial proximity of homologous regions of centromere can facilitate genomic rearrangements. They demonstrate it by performing genome wide synteny analysis of Ct using Ca as reference genome. Results show that all the centromeres in Ct are either present at an interchromosomal synteny breakpoint (ICSB) or near an ICSB (except CEN6). Authors have also showed presence of HIR associated putative centromeres in species closely related to Ct i.e. C. sojae and C. viswanathii which is lost in Ca and C. dubliniensis (closely related species to Ca). Overall, this study proposes that close spatial proximity of homologous regions of centromeres in Ct led to inter-centromeric translocation events which possibly led to formation of evolutionary new centromeres on unique DNA sequences in Ca.Strengths:1) Very coherently demonstrated a possible mechanism for evolution of HIR-associated centromeres to centromere formation on unique DNA sequences in two closely related species Ct and Ca.

We thank the reviewer for encouraging words.

2) Generated chromosome level assembly of Ct and fragmented level assembly of C. sojae. This will act as an important resource for the scientific community.

Thanks for pointing out one of the critical aspects of this study. Indeed, our primary motivation was to generate a chromosome-level genome assembly of *Candida tropicalis,* an emerging medically relevant but poorly studied human pathogen. We hope that the genomic resources made available for the scientific community through this work will facilitate future studies.

3) In Assembly2020, author have identified various key structural regions like long copy number variations, long-track loss of heterozygosity and heterozygous translocation events providing a wholistic genomic map.

We appreciate that the reviewer highlighted this point. In fact, as a follow-up study for the next paper, we are trying to understand the importance of some of these genomic features in the drug resistance of the organism.

Minor Comments:4) Title: Please correct: “inter-centromeric” instead of “intercentromeric”

We have made this change throughout the manuscript.

5) Please share the codes used to run the programs or share the GitHub link for the codes.

We have uploaded the codes written by us associated with 3C-seq data analysis to a Github repository and added the link in the revised manuscript (https://github.com/YaoChen/candida-tropicalis-analysis). For codes from publicly available programs/software, we have cited the references and mentioned the details of parameters in the Materials and method section.

6) Figure 1—figure supplement 2C-G: Please explain the assay used to determine aneuploidy in more details. The rationale and the explanation are not provided anywhere in the paper or in the supplementary information. What is the purpose of deleting SCH9?

We thank the reviewer for the suggestion. We have now described in detail the rationale and explained the steps followed to generate monosomic strains (CtKG101 – 105) (see Materials and methods). The deletion of *SCH9* alleles increases the rate of chromosome loss in *C. albicans* (Varshney et al., 2015). To exploit this property of *sch9* mutants, we deleted both copies of *SCH9* in *C. tropicalis* and created a reporter strain (CtKG002) in the *sch9* mutant background to assay for loss of Chr5. In this reporter strain, the short arm (5ʹ end) of one of the two homologs of Chr5 has been marked with *URA3*, and the long arm (3ʹ end) carries the *MTL***a** or *MTLα* locus in such a way that two distinct alleles present on two homologs. *URA3* and *MTL* loci are thus unlinked to each other. Next, we plated different dilutions of CtKG002 cells on plates containing 5-Fluoroorotic Acid (5-FOA) media to perform a counter selection of cells, which have lost the *URA3* marker and grown as 5-FOA^R^ single colonies. We confirmed the simultaneous loss of *URA3* marker and one of the MTL alleles. Strains showing concomitant loss of these two unlinked loci must have lost an entire homolog of Chr5 and, therefore, are monosomic for Chr5. These strains (2n-1) that lacked one homolog of Chr5 were used to confirm the presence of heterozygosity of orphan haplotigs (OHs) of CtChr5.

7) Explain Figure 1—figure supplement 2E in more details.

We have now explained Figure 1—figure supplement 2E in more detail.

8) Figure 1—figure supplement 2D: Shouldn't the higher band be marked as 4580bp and lower band be marked as 3312bp?

We thank the reviewer for pointing out this error. We have corrected Figure 1—figure supplement 2D in the revised manuscript.

9) Results paragraph one: Introduce what are orphan haplotigs in the text like we suspect orphan haplotigs to be regions of aneuploidy in the Ct genome.

We thank the reviewer for the suggestion. In the revised manuscript, we have defined “orphan haplotigs” as “suspected heterozygous loci in the diploid genome of *C. tropicalis”*.

10) Subsection “Centromere and telomere proximal loci are hotspots for complextranslocations”: Any comments on why Cen6 did not have any inter-centromeric translocation? Anything special about its structure or location of any conserved element or essential genes near the centromere that might prevent gene rearrangement event?

There is no explanation of why *CtCEN6* did not show any inter-centromeric translocation with respect to the *C. albicans* genome. However, synteny analysis of ORFs proximal to *CEN6* of *C. tropicalis* (*CtCEN6*) reveals that three *CtCEN6*-associated ORFs are absent, but the other flanking ORFs are present in the *C. albicans*. We have mentioned these points in the revised manuscript. It is important to note that by using the *C. tropicalis* genome as the reference, all centromeres of *C. albicans*, except *CaCEN2*, were found to be associated with ICSBs (Figure 3—figure supplement 1B). Taking together, centromeres of both these species are found to be associated with chromosomal translocations.

11) Paragraph three subsection “Rapid transition in the centromere type within the members of the CUG-Ser1 clade”: It is not clear what species were used for this analysis. Was the IR-motif recognized in inverted repeats of all four species Ct, C. sojae, C. viswanathii, and Ca or just the first three? From the subsequent analysis, it is clear that IR-motifs are enriched in near centromeric regions of Ct, C. sojae, and C. viswanathii. If Ca was used for analysis, then what is the distribution of IR-motif in Ca.

We have now clearly explained the method used to identify the IR-motifs. Briefly, the centromeric IR sequences from *C. tropicalis* and those from putative centromeres of *C. sojae* and *C. viswanathii* were analysed to identify presence of any conserved motifs using motif discovery tool MEME (Bailey, et al., 2009). Having identified the reported IR motif, we scanned the chromosomes using FIMO (Bailey, et al., 2009). Subsequently, we compared the average IR-motif density at centromere with that of the genomic average in all the *Candida* species studied here. We do not see an enrichment of IR-motifs in *C. albicans* or *C. dubliniensis* centromeres compared to the genome average while the IR-motif is at least ~10 fold enriched in *C. tropicalis, C. sojae, C. viswanathii* centromeres (Figure 4C).

12) It was interesting to see the conservation of IR-motif across species. Any comments on what can be the significance of this conserved motif? Does it have any close resemblance to any known protein binding sites?

Previously, we have tested the contribution of the DNA sequence of the centromeric IRs in *C. tropicalis* in conferring mitotic stability and *de novo* CENP-A loading functions by replacing the native IRs with IR sequences of same length from *C. albicans CEN5* in the plasmid context. This study clearly showed the inability of Ca*CEN5* IR in complementing *the CEN* function (Chatterjee et al., 2016). Our observation that the IR-motifs are absent in Ca*CEN5* suggests these motifs may be required for *de novo* centromere establishment in *C. tropicalis*. It is possible that these motifs serve as binding sites for a specific protein, centromere-specific enrichment of which contributes to *de novoCEN* function. However, the exact mechanism remains unexplored and can be addressed by performing more specific experiments, which are beyond the scope of this study.

Reviewer #1 (Significance (Required)):Candida are pathogenic yeast that presents a growing threat to human health. The diversity of species contributes to the difficulty in treatment. Gapless assembly will further aid in research and allowed the authors to delve further into the evolutionary history of genome arrangements.

Thank you for the positive comments and encouraging words of appreciation.

Reviewer #2 (Evidence, reproducibility and clarity (Required)):Summary:This work demonstrates whole genome assembly of Candida tropicalis, which is a related species with well-studied Candidida albicans. While centromeres are formed on unique DNA sequences in C. albicans, C. tropicalis possesses homogenized inverted repeat (HIR) associated centromeres. Comparison of whole genome sequences of both species, authors found evidences for intercentromeric translocations in the common ancestor of both species. They also predicted that HIR sequences were lost during intercentromeric translocations in lineage of C. albicans and acquired evolutionary new centromere (ENC) in this species. Consistent with this idea, they found HIR associated centromeres in C. parapsilosis, C. sojae, and C. viswanathii, but not C. dubliniensis as well as C. albicans.Major comments:1) Genome sequence and informatics analyses have been done well. The methodology in this paper is reliable. Identification of centromeres of C. tropicalis at ICSB is interesting observation.

We thank the reviewer for encouraging comments.

2) However, additional data for centromere identification in C. parapsilosis, C. sojae, and C. viswanathii may be needed. CENP-A ChIP experiments will strengthen quality of their conclusion.

We thank the reviewer for this suggestion. Coincidentally, the genomic coordinates of the putative centromeres reported in this study match perfectly with the genomic locations of CENP-A-rich centromeres in *C. parapsilosis* strain CLIB214 identified in a recent study (Ola et al., 2020). While we agree that an additional CENP-A ChIP experiment can validate the putative centromeres in the other two species, neither antibodies against CENP-A^Cse4^ nor epitope-tagged CENP-A^Cse4^ expressing strains of these species are available. Once the transformation protocol of these two species is established, such experiments can be done in the future. In addition, common features such (a) the conserved IR-associated structure (Figure 4A, Figure 4—figure supplement 1A), (b) overall conservation in DNA sequence compared to random genomic loci (Figure 4A, Figure 4—figure supplement 2A-B), (c) the presence of conserved motifs (Figure 4D-E, Figure 4—figure supplement 2C-D) with same direction (Figure S10F), and (d) fully or partially conserved gene synteny across the centromeres of *C. tropicalis* with the putative centromeres of *C. sojae* (Figure 4—figure supplement 1B, D), and *C. viswanathii* (Figure 4—figure supplement 1C, E) indicate that these HIRassociated loci are probably authentic centromeres.

3) Furthermore, if addition of C. tropicalis cen sequence to the replicating plasmid facilitates *de novo* CENP-A assembly, centromeres of C. parapsilosis, C. sojae, and C. viswanathii may also have similar activity. This would suggest that HIR may be a genetic element for centromere specification. If these experiments can be done, it would be good.

We thank the reviewer for this suggestion. Previously, we have tested the role of centromeric IRs in *de novo* CENP-A recruitment on the centromeric plasmid of *C. tropicalis* (Chatterjee et al., 2016). A similar experiment can be performed by cloning the HIRassociated centromere DNA of *C. sojae* and *C. viswanathii* and assaying for improvement of mitotic stability and *de novo* CENP-A loading. We would like to try these experiments by constructing strain expressing an epitope-tagged CENP-A^Cse4^, once a transformation protocol of these two species can be standardized.

4) Does Cen6 in C. tropicalis have a HIR sequence? As it is a bit unclear, please state this point clearly in the revised version.

Our previous study reported that all seven centromeres are HIR-associated, which is stated in the original submission: “Strikingly, all seven centromeres of another CUG-Ser1 clade species *C. tropicalis*…highly identical to each other.” Our current analysis also showed that all seven centromeres of *C. tropicalis,* including *CEN6,* carry homogenized inverted repeat (Figure 4A) and contain the IR-motifs (Figure 4D-E). However, the central core of *CEN6* harbors two ORFs, unlike the ORF-free CCs of the other six centromeres. We have mentioned these points in the revised version (Introduction).

5) If their intercentromeric translocation theory is correct, how HIR of cen 7 in C. albicans (corresponding Cen6 in C. tropicalis) are lost? Furthermore, Cen6 in C. tropicalis also makes a cluster with other centromeres, so why Cen6 was escaped from intercentromeric translocation? As I understand that it is hard to explain these points experimentally, please discuss it in the revised version.

Although it appears that *CtCEN6* escaped inter-centromeric translocations, synteny analysis suggested that a chromosomal region carrying three consecutive *CtCEN6*proximal ORFs was lost in the *C. albicans* genome. This suggests that CtCEN6 had gone through chromosomal rearrangements. This can be explained by a double-stranded DNA break at *CtCEN6* followed by a fusion of broken ends. This event might have led to the loss of HIRs and the emergence of an ENC on Chr7 in *C. albicans*. We speculate that more intracentromeric rearrangements might have taken place during the transition from HIRassociated centromeres to unique repeat-less ENCs.

Reviewer #2 (Significance (Required)):This is a solid work and should be published in an appropriate journal. However, this work is on centromere evolution within Candida species, people outside the filed may not have wide interests. Even in centromere research field, it is hard to evaluate generality of this finding. If similar events happen in other species outside fungi, this would be interesting widely.

Thanks for the appreciation. With centromere features are known in more than 50 species, fungal centromeres are not only well-characterized but also show a wide diversity of centromere features found in other forms of life such as in plants and animals. Strikingly, the only feature that remained conserved among all fungal centromeres is their spatial clustering (Guin, Sreekumar, Sanyal, *in press*. Annual Review of Microbiology). In this work, we provide mechanistic evidence to support how spatial clustering favors centromere type transition, which may be found in animal and plant systems as centromeres are shown to be one of the most rapidly evolving loci in all domains of life (Henikoff et al., 2001; Padmanabhan et al., 2008).

We found genomic evidence of inter-centromeric translocation, which may have led to the loss of HIR-associated ancestral type and emergence of evolutionary new centromeres in *C. albicans*. However, even during such dramatic karyotype reorganization, the clustering of centromeres remained unperturbed (Figure 5C). This observation indicates that centromere clustering in fungi is a matter of cardinal importance. Based on our results (Figures 3 and Figure 3—figure supplement 1), we proposed that spatial proximity and DNA sequence homology aided karyotypic rearrangements and possibly aided speciation events in CUG-Ser1 clade. This conserved principle of spatial proximity and DNA sequence homology favoring recombination holds true among multiple domains of life and can be related to other well-studied phenomena. interchromosomal contacts between chromosome pairs have been correlated with the number of translocation events in both naturally occurring populations and experimentally induced mammalian cells (Arsuaga et al., 2004; Bickmore and Teague, 2002; Branco and Pombo, 2006; Canela et al., 2017; Engreitz et al., 2012; Hlatky et al., 2002; Holley et al., 2002; Klein et al., 2011; Roukos et al., 2013; Zhang et al., 2012). It has been suggested that contacts between various chromosomal territories as well as their relative positions in the nucleus influence the sites and frequency of translocation events both in flies and mammals (Roukos et al., 2013; Soutoglou and Misteli, 2008). One such well-studied translocation events, Robertsonian translocation (RT), involving the fusion between arms of two different chromosomes near the centromere, is the most frequently detected chromosomal abnormality in humans (Therman, Susman et al., 1989). The occurrence of RT was first reported in grasshoppers (Robertson 1916) and subsequently been implicated in karyotype evolution in humans (Therman, Susman et al., 1989), mice (Castiglia and Capanna 2002, Dumas and Britton-Davidian 2002), and wheat (Friebe, Zhang et al., 2005) among others. Although significantly different from centromere clustering in fungi, cytological and Hi-C based evidence of spatial proximity (reviewed in Muller, Gil et al., 2019; Imakaev et al., 2012) among the repeat-associated centromere DNA sequences (Kalitsis, Griffiths et al., 2006) in humans, mice and wheat supports a possibility that RT may have been guided by spatial proximity. Similarly, chromoplexy, involving a series of translocations among multiple chromosomes without alteration in copy number, was identified in prostate cancers (Baca, Prandi et al., 2013; Zhang, Leibowitz et al., 2013). Although fine mapping of translocation events at the repetitive regions in human cancer cells becomes difficult, growing evidence suggests that such events are associated with the formation of micronuclei (Crasta, Ganem et al., 2012). This further supports the idea that the spatial genome organization may influence chromoplexy (Meaburn, Misteli et al., 2007). Therefore, we strongly believe that evolutionary implications of our observations will be of interest to a broader group of researchers studying not only centromere biology but also mechanisms of genome evolution and speciation in fungi and beyond.

Reviewer #3 (Evidence, reproducibility and clarity (Required)):This study uses multiple different technologies to improve the genome assembly of Ct. They also end up resolving haplotype-specific differences, copy number variations, a translocation, and an LoH event while doing so. The interesting hypothesis relates to differences centromere formation of different Candida species mainly focusing on the wellstudied Ca genome and the newly generated Ct assembly here. In my opinion, this is a complete piece of work with one clear deliverable (new assembly) and several interesting hypotheses, some of which would require further studies for a definitive conclusion.

We thank the reviewer for encouraging remarks.

I have only a few major concerns but the manuscript needs a good grammar and consistency check before publication.

We have tried to the best of our ability to check for grammatical errors and consistency throughout the manuscript.

1) Could the authors generate 3Cseq data for one other species from the CUG-Ser1 clade (C. sojae) to show the expected centromere locations (HIRs) cluster in 3D?

The species studied to date in subphylum Saccharomycotina show clustered centromeres at all stages of the cell cycle (reviewed in Muller et al., 2019). Among the members of the CUG-Ser1 clade, clustering of the centromere-kinetochore complex was first studied in *C. albicans* (Sanyal and Carbon, 2002) and was later confirmed by Hi-C analysis (Burrack LS et al., 2016). In fact, the centromeres of *C. albicans* share highly enriched transchromosomal contacts (Sreekumar L. et al., 2019). This property of centromere clustering in fungi was used to develop analytical techniques for prediction of centromere location using genome-wide contact probability data (Varoquaux, N. et al., 2015), which showed clustering of centromeres in the CUG-Ser1 clade member *Scheffersomyces stipitis*. Similarly, Hi-C data revealed clustering of centromeres in another CUG-Ser1 clade member *Debaromyces hansenii* (Marie-Nelly H. et al., 2014). Based on these known facts, we expect similar clustering of centromeres in *C. sojae.* This can be tested by sub-cellular localization of CENP-A^Cse4^ and ChIP-qPCR to validate CENP-A enrichment on the HIR-associated putative centromere loci. However, once the transformation protocol in this organism is established, it will be possible for us to perform this experiment. We also plan to perform Hi-C experiments to improve the genome assembly and observe genomic features of *C. sojae* in a future study.

2) The link to chromothripsis (defined in its originally proposed form) is not clear to me. This has to be either elaborated more or just removed. The events mentioned are consistent with multiple interchromosomal translocations, not shattering of a chromosome.

In the initial draft of our manuscript, we used the phrase “chromothripsis-like event” to explain the inter-centromeric translocations observed in the last common ancestor of *C. albicans* and *C. tropicalis.* In this revised version, we have removed the same.

3) Results from Figure 3C and 3D need to be consolidated. I am not sure how ICSB density can be higher at CP compared to TP (Figure 3C) but yet the lengths of orthoblocks are in general shorter for TP compared to CP (Figure 3D).

Figure 3C shows the ICSB density on six chromosomes of *C. tropicalis* (except Chr6, which does not carry any ICSB) as a function of the distance from the centromere. On the other hand, Figure 3D compares the length of orthoblocks present at centromere-proximal (CP), centromere-distal (CD), and telomere-proximal (TP) zones. These two genomic features are independent of each other. For example, the ICSB density at CP is higher than TP (Figure 3C), but the lengths of orthoblocks are shorter at TP than CP (Figure 3D). This is due to the clustering of majority (28/39) of ICSBs at CP, while few (7/39) smaller blocks are present at TP.

4) Several other narrower spikes in 3C coverage for chr1 and chr2 (Figure 1D). How were these determined to not be duplicated regions? The methodological details of how these decisions were made are missing.

We thank the reviewer for the suggestion. Now we have exploited a more comprehensive approach to detect CNVs across the genome using a published tool CNAtra (Khalil et al., 2020). The estimated copy numbers are computed for each region, and those with estimated copy number >2.5 are considered as duplicated regions. Details of the CNV detection method have been described in Materials and methods.

5) How can the 3Cseq data be generated using HindIII for digestion and can be binned at 5bp bins or 2kb bins?

We performed the 3C-seq experiment using DpnII and not HindIII (see Materials and methods). To determine the copy number of orphan haplotigs, paired-end 3C-seq reads were mapped to orphan haplotigs and a control locus (Figure 1—figure supplement 2A) using paired-end alignment mode. The reads were mapped per 5-bp bin and normalized to per million mapped reads. These values were then used to show that specific regions of the orphan haplotigs are present in one copy, while the control locus is present in two copies (Figure 1—figure supplement 2A – B).

Minor:6) Figure 1 caption mentions B in place of C, and vice versa, compared to the figure.

We have corrected it in the revised manuscript.

7) Figure 2C and other heatmaps. The exact values of the color scale need to be reported rather than high vs low.

We have now modified the figures to show the exact values of the color scale.

- scatter-pot – scatter-plot, conseqence – consequence

They have been corrected in the revised manuscript.

8) Replace roman numerals in Figure 3C with distance ranges, it is confusing. Figure 3D in the caption is marked as "E.", replace it with D.

We have now replaced the Roman numerals with the distance range. Figure 3D in the caption was marked as "E.," now we have replaced it with D.

9) How large are the genomic regions in Figure 4E? This needs a scale bar to show the size in kb

We have now mentioned the length of each of the locus in the modified Figure 4E and Figure 4—figure supplement 2D.

10) Figure 5 caption: repeat associated-associated.

We have corrected this error.

Reviewer #3 (Significance (Required)):

- *A new chr-level assembly for C. tropicalis*

- Hypotheses and some supporting information about the evolution of centromere formation in related species.- Important for yeast biologists- I have expertise in computational analysis of conformation capture data and analysis of such data in related species

Thanks for highlighting the significant findings of this study.